# ARID1B maintains mesenchymal stem cell quiescence via inhibition of BCL11B-mediated non-canonical Activin signaling

Mingyi Zhang[1], Tingwei Guo[1], Fei Pei [1], Jifan Feng[1], Junjun Jing[1], Jian Xu [1], Takahiko Yamada [1], Thach-Vu Ho[1], Jiahui Du[1], Prerna Sehgal[1] & Yang Chai [1] ✉

*ARID1B* haploinsufficiency in humans causes Coffin-Siris syndrome, associated with developmental delay, facial dysmorphism, and intellectual disability. The role of ARID1B has been widely studied in neuronal development, but whether it also regulates stem cells remains unknown. Here, we employ scRNA-seq and scATAC-seq to dissect the regulatory functions and mechanisms of ARID1B within mesenchymal stem cells (MSCs) using the mouse incisor model. We reveal that loss of *Arid1b* in the GLI1+ MSC lineage disturbs MSCs' quiescence and leads to their proliferation due to the ectopic activation of non-canonical Activin signaling via p-ERK. Furthermore, loss of *Arid1b* upregulates *Bcl11b*, which encodes a BAF complex subunit that modulates non-canonical Activin signaling by directly regulating the expression of activin A subunit, *Inhba*. Reduction of *Bcl11b* or non-canonical Activin signaling restores the MSC population in *Arid1b* mutant mice. Notably, we have identified that ARID1B suppresses *Bcl11b* expression via specific binding to its third intron, unveiling the direct inter-regulatory interactions among BAF subunits in MSCs. Our results demonstrate the vital role of ARID1B as an epigenetic modifier in maintaining MSC homeostasis and reveal its intricate mechanistic regulatory network in vivo, providing novel insights into the linkage between chromatin remodeling and stem cell fate determination.

Chromatin remodeling complexes play crucial roles in the epigenetic regulation of transcription, DNA damage repair, and DNA unwinding by modifying chromatin accessibility in a fine-tuned manner. The BRG1/BRM-associated factor (BAF) complex, also known as the mammalian switch/sucrose non-fermenting (SWI/SNF) complex, is a chromatin remodeler that plays an essential role in regulating organ development, tissue homeostasis, disease development, and cancer biology[1–7]. *ARID1B* encodes the protein AT-rich interactive domain 1B (ARID1B), the signature subunit of the BAF complex, which is mutually exclusive with ARID1A[8]. In humans, haploinsufficiency of *ARID1B* is associated with neurodevelopmental disorders and syndromic/non-syndromic intellectual disabilities, including Coffin-Siris syndrome and HHID syndrome[9,10], and *ARID1B* mutations have been identified in

different human cancers[11,12]. Studies conducted on mice have revealed the roles of ARID1B in neurodevelopment[13,14] and cranial neural crest cell formation[15–17]. Despite the increasing researches on ARID1B, its roles and regulatory mechanisms in cell fate commitment, stem cell homeostasis, and disease/cancer biology remain largely unknown. It is necessary to explore the functional regulatory network of the epigenetic regulator ARID1B and unravel its molecular mechanisms related to stem cell function to provide essential insights into its roles in regulating tissue homeostasis and diseases.

Stem cells are essential players in organ development and maintaining tissue homeostasis, possessing the unique ability to self-renew and differentiate into multiple cell types. The mouse incisor has been established as an ideal animal model for studying the self-renewal and

[1]Center for Craniofacial Molecular Biology, University of Southern California, Los Angeles, CA 90033, USA. ✉e-mail: ychai@usc.edu

 1

differentiation of mesenchymal stem cells (MSCs) as they contribute to tissue homeostasis and injury repair. Mouse incisor's proximal end harbors MSCs that support its continuous growth throughout life[18,19]. Our previous studies have shown that *Gli1* is an in vivo MSC marker, and GLI1+ cell population is located near the neurovascular bundle (NVB) in the proximal region of the incisor; these cells can give rise to transit-amplifying cells (TACs) that eventually populate the entire dental mesenchyme[19–22]. The differentiation axis of cells in the mouse incisor has a specific proximal-to-distal orientation, such that one finds MSCs most proximally, then TACs, which differentiate into odontoblasts and dental pulp cells in the more distal regions[19,23]. This spatial organization makes the mouse incisor an ideal system for studying the gene regulatory roles of MSCs and tissue homeostasis.

Similar to other types of stem cell, the homeostasis of MSCs in the mouse incisor is governed by a complex and interconnected signaling regulatory network[19,24–29]. Notably, TGF-β signaling has been extensively studied in various cellular contexts; however, the Activin signaling pathway, which belongs to the TGF-β superfamily, is less known. Activin signaling's potential interplay with the MSC population and its contributions to MSC regulation and fate determination is still elusive. Understanding the involvement of Activin signaling and the crosstalk between its ligands and receptors in the maintenance of MSC-mediated tissue homeostasis is crucial for unraveling the underlying mechanisms governing MSC self-renewal, differentiation, and overall functional properties.

In the epigenetic regulation of MSCs, a PRC1-WNT regulatory network has been identified as governing reciprocal interactions between MSCs and TACs[24]. In this study, we unveiled the vital role and mechanistic regulatory network of another epigenetic regulator, ARID1B, in maintaining MSC quiescence and tissue homeostasis in vivo. Through scRNA-seq and scATAC-seq analysis, we identified that loss of *Arid1b* disturbs MSCs' quiescence and induces their proliferation. ARID1B directly represses the expression of *Bcl11b* by binding to its third intron to maintain MSC homeostasis. Following the loss of *Arid1b*, the upregulated BCL11B serves as a mediator that directly regulates the expression of *Inhba*, which encodes activin A ligand of the Activin signaling pathway, which in turn ectopically activates non-canonical Activin signaling (p-ERK) in the MSC region. Significantly, we verified the functional significance of BCL11B and non-canonical Activin signaling downstream of ARID1B in preserving MSC homeostasis. Our results reveal the spatiotemporal dynamics of an epigenetic regulatory network involving ARID1B, BCL11B, and non-canonical Activin signaling, which form a cascade that contributes to the maintenance of MSCs in tissue homeostasis.

## Results

### ARID1B plays an essential role in regulating adult mouse incisor growth and tissue homeostasis

To investigate the role of ARID1B in regulating MSC fate commitment and mesenchymal tissue homeostasis, we evaluated the expression pattern of ARID1B in the proximal region of the mouse incisor. We found that ARID1B is widely expressed in the dental mesenchyme near the NVB where MSCs reside, as well as in odontoblasts, dental pulp cells, and epithelial cells, but less in the TAC region and pre-odontoblasts (Fig. 1a, b). Previous study has shown that GLI1+ cells are MSCs surrounding the NVB[19]. To find out whether ARID1B is expressed in these GLI1+ MSCs, we co-stained GLI1 and ARID1B and found that ARID1B+ cells in the NVB region overlap with a sub-population of GLI1+ cells (Fig. 1c, d). Thus, we hypothesized that ARID1B plays a role in regulating MSC commitment and tissue homeostasis in the adult mouse incisor.

GLI1+ MSCs located surrounding the NVB support the mouse incisor's growth and replenishment throughout the lifespan. Using the *Gli1-CreER* line, we generated *Gli1-CreER;Arid1b^{fl/fl}* mice, in which *Arid1b* was inactivated in the GLI1+ lineage after tamoxifen induction at one month of age, and confirmed that *Arid1b* was effectively deleted from

the incisor mesenchymal and epithelial cells (Supplementary Fig. 1a–d). To evaluate the impact of the loss of *Arid1b* in MSCs on the mouse incisor growth, we first performed an incisor growth assay by comparing the movement of a notch made in the incisor enamel above the gingival margin in both control and *Arid1b* mutant mice. We found that the growth rate was significantly slower in *Arid1b* mutant mice than in the control across all measurement time points (Fig. 1e–k). This result indicated that the loss of *Arid1b* in the GLI1+ lineage impairs the adult mouse incisor growth.

Next, we assessed the long-term impact following the loss of *Arid1b*. At 2 months post-tamoxifen induction, the dental pulp cavity was narrower in *Gli1-CreER;Arid1b^{fl/fl}* mice than in the control, as shown by microCT (Supplementary Fig. 1e, h). Histologically, the polarization of odontoblasts (Supplementary Fig. 1f, i) and the expression of odontoblast differentiation marker *Dspp* (Supplementary Fig. 1g, j) were initiated more proximally to the cervical loop in *Arid1b* mutant mice than in the control mice. The cervical loop also appeared to be smaller in the *Arid1b* mutant mice. The odontoblast alignment was affected, and the dentin appeared thicker in the *Arid1b* mutant mice. Moreover, the phenotype of the *Arid1b* mutant mice became more severe at 3 months post-tamoxifen induction, with a limited dental pulp cavity and stacked dentin at the proximal end of the incisor (Fig. 1l–o, r). The odontoblasts were well aligned, and their differentiation was marked by the organized alignment of nuclei along the basement membrane in the control; in contrast, the organization of the odontoblasts was abnormal, and the nuclei were in the opposite position in *Arid1b* mutant mice. The odontoblasts were premature in the proximal region in the *Arid1b* mutant mice, as confirmed by *Dspp* marker staining (Fig. 1p, q, s). These results indicated that ARID1B plays a role in maintaining adult mouse incisor tissue homeostasis.

Furthermore, to investigate the cellular changes underlying the reduced growth rate and abnormal dentin formation in *Arid1b* mutant incisors, we evaluated the potential of TACs to differentiate into odontoblasts after the loss of *Arid1b*. We labeled TACs in the DNA synthesis phase using EdU injection and harvested the tissue 48 h later to assess the TAC differentiation. The overlap between *Dspp*+ odontoblasts and EdU-labeled cells represented the TAC differentiation ability, and the overlap length indicated the migration rate of these differentiated cells during the preceding 48 h. We observed a reduced overlap length of EdU+/*Dspp*+ cells in *Arid1b* mutant mice compared to controls at 1 week post-induction, indicating that loss of *Arid1b* caused the compromised migration rate of the differentiated TACs (Supplementary Fig. 1k–n, q). To understand the cause of abnormal dentin formation, we administered calcein and alizarin red S dual fluorescence injections at different time points to dynamically compare the odontoblast migration rate and dentin deposition rate. Using this approach, the fluorescence precipitation represents dentin formation at the time of injection. We compared the odontoblast migration length and dentin deposition depth over a period of 5 days. Statistical analysis revealed a significantly shorter odontoblast migration length in *Arid1b* mutant mice compared to controls, while there was no significant difference in the depth of dentin deposition (Supplementary Fig. 1o–p, r). These findings indicated that the loss of *Arid1b* impairs the odontoblast migration rate, leading to abnormal stacked dentin.

GLI1+ cells contribute to the mesenchymal and epithelial lineages in the mouse incisor[19] and ARID1B was effectively deleted in both tissues in the *Gli1-CreER;Arid1b^{fl/fl}* mice. To clarify whether the loss of *Arid1b* in the dental epithelium has any effect on incisor tissue homeostasis, we used the *Sox2-CreER* line to specifically delete *Arid1b* in the epithelial lineage by generating *Sox2-CreER;Arid1b^{fl/fl}* mice. Again, we induced Cre activity with tamoxifen at 1 month of age and harvested the incisor at 2 months post-induction. There were no signs of either premature odontoblasts or a differentiation defect in *Sox2-CreER;Arid1b^{fl/fl}* mice as compared to the control (Supplementary Fig. 2a–f). This result indicated that the loss of *Arid1b* specifically in the

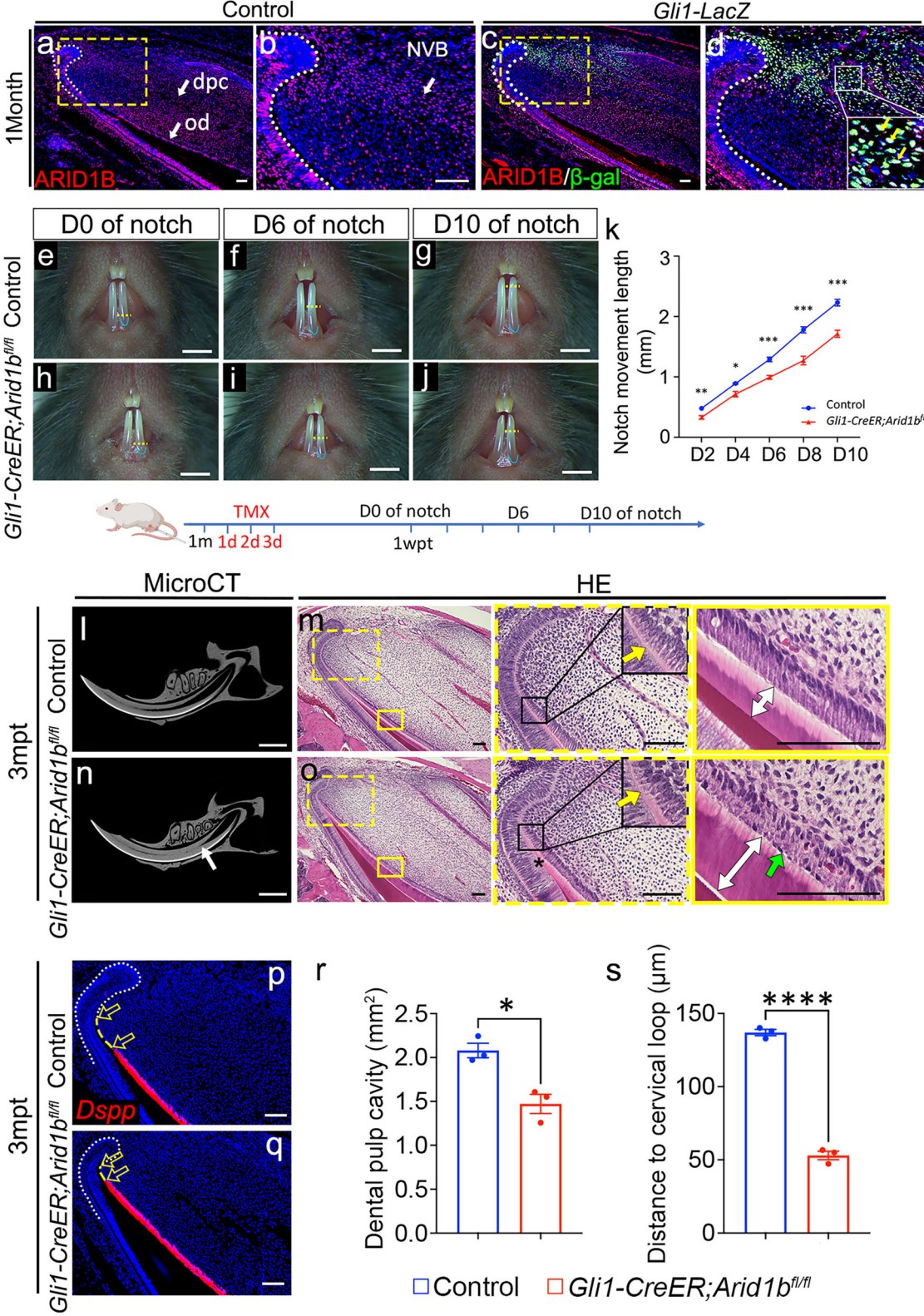

dental mesenchyme impairs the incisor growth and tissue homeostasis.

## Loss of *Arid1b* disturbs MSCs' quiescence and leads to their proliferation

The continuous growth of the mouse incisor is induced by the progression of MSCs giving rise to TACs, along with proliferation and differentiation in the proximal region. Due to the overlap between ARID1B+ cells and a subpopulation of GLI1+ cells in the proximal region, we investigated the potential impact of *Arid1b* loss on GLI1+ MSCs. We generated *Gli1-CreER;Arid1b^{fl/fl};Gli1-LacZ* mice to compare the numbers of GLI1+ cells in the incisors of these mutants and *Gli1-LacZ* control mice. At 5 days post-induction, the number of GLI1+ cells was significantly reduced in the *Gli1-CreER;Arid1b^{fl/fl};Gli1-LacZ* mice compared to *Gli1-LacZ* mice (Fig. 2a–e). These MSCs are quiescent cells that undergo slow-cycling self-renewal and reside in the proximal

**Fig. 1 | Loss of *Arid1b* in GLI1+ lineage impairs mouse incisor growth and tissue homeotasis. a–d** Immunostaining of ARID1B (red) in control (**a**, **b**) mouse incisor and co-immunostaining of ARID1B (red)/β-galactosidase (β-gal) (green) in *Gli1-LacZ* mouse incisor (**c**, **d**). **b**, **d** represent the high-magnification image of the box in (**a**, **c**). White dotted lines outline the cervical loop. White arrows indicate ARID1B+ cells. Yellow arrows indicate the ARID1B+/GLI1+ cells. **e–j** Notch movement assay in control (**e–g**) and *Gli1-CreER;Arid1b^{fl/fl}* (**h–j**) mice. Yellow dotted lines show the notch position. Blue lines show the gingival margin. TMX, tamoxifen; wpt, week post-tamoxifen injection. Illustration below (**e–k**) created with BioRender.com released under a Creative Commons Attribution-NonCommercial-NoDerivs 4.0 Internatioanl license. **k** Quantification of the notch movement every other day from D2 to D10. Data are mean ± SEM, control *n* = 5, *Gli1-CreER;Arid1b^{fl/fl}* *n* = 7, unpaired two-tailed Student's t-test. D2, *p* = 0.0083; D4, *p* = 0.0103; D6, *p* = 0.0001; D8, *p* = 0.0003; D10, *p* = 0.0001. **l–o** MicroCT and HE staining of control (**l**, **m**) and *Gli1-CreER;Arid1b^{fl/fl}* (**n**, **o**) mouse incisors at 3 months after tamoxifen induction.

**l**, **n** MicroCT of control (**l**) and *Gli1-CreER;Arid1b^{fl/fl}* (**n**) mouse incisors. White arrow indicates the narrowed dental pulp. **m**, **o** HE staining of control (**m**) and *Gli1-CreER;Arid1b^{fl/fl}* (**o**) mouse incisors. Yellow arrows indicate the initiation of odontoblast polarization. Green arrow indicates the disordered alignment of odontoblasts. White two-way arrows indicate the dentin thickness. Black asterisk indicates stacked and distorted dentin. Boxes in m and o are shown at higher magnification on the right. **p**, **q** *Dspp* (red) in situ hybridization in control (**p**) and *Gli1-CreER;Arid1b^{fl/fl}* (**q**) mouse incisors. White dotted lines outline the cervical loop. Yellow dotted lines show the cervical loop bending point to the odontoblast initiation distance. Unfilled arrows indicate the distance between the yellow dotted lines. *n* = 3. **r** Quantification of the dental pulp cavity. Data are mean ± SEM, *n* = 3, unpaired two-tailed Student's t-test. *p* = 0.0113. **s** Quantification of the distance of *Dspp*+ cells to the cervical loop. Data are mean ± SEM, *n* = 3, unpaired two-tailed Student's *t*-test. *p* < 0.0001. Source data are provided as a Source Data file. Scale bars: 2 mm (**e–j**, **l**, **n**); 100 μm (other images).

region of the mouse incisor. Thus, we investigated the impact of ARID1B on MSCs' quiescence based on their label-retaining ability. Since incisor mesenchyme turnover takes about 1 month[19], we injected control and *Gli1-CreER;Arid1b^{fl/fl}* mice with EdU for a 1-month period beginning from postnatal day 5 and analyzed the cells after another month. In this case, the label retaining cells (LRCs) detected by EdU staining would be the quiescent cells. We found the number of LRCs was significantly reduced in *Gli1-CreER;Arid1b^{fl/fl}* mice compared to controls (Fig. 2f–j), confirming that loss of *Arid1b* impairs mouse incisor MSC quiescence.

To understand the reason for the reduction in GLI1+ MSCs following the loss of *Arid1b*, we first conducted TUNEL assay to compare apoptosis levels between control and *Arid1b* mutant mice. The results showed that there was no obvious increase in apoptosis at 5 days post-induction, suggesting that the decrease in MSCs was not caused by an increase in cell death (Supplementary Fig. 2g–j). To further investigate the underlying causes for MSC changes, we performed single-cell RNA sequencing (scRNA-seq) on the proximal incisor tissue from both control and *Gli1-CreER;Arid1b^{fl/fl}* mice 4 days post-tamoxifen induction. We integrated the scRNA-seq data from the control (9210 cells) and *Gli1-CreER;Arid1b^{fl/fl}* (8915 cells) mice, and identified a large group of dental mesenchymal cells along with other cell types via unsupervised clustering and marker analysis (Supplementary Fig. 3a, b). We then focused on the dental mesenchymal cells and subclustered them into five distinct populations (Fig. 2k). We identified the specific markers for each cluster and validated their expression patterns in the mouse incisor (Supplementary Fig. 3c–e). To investigate the effects of *Arid1b* loss at the single-cell level across different cell clusters, we compared the relative percentage of cells in each cluster between control and *Arid1b* mutant samples. We observed a decrease in the number of MSCs in the *Arid1b* mutant sample, consistent with the in vivo marker analysis, alongside an increase in the TAC number (Fig. 2l). The comparison at the single-cell level suggested that the loss of *Arid1b* may impact MSC quiescence by inducing MSC proliferation.

To validate our hypothesis, we utilized scRNA-seq data to construct a computational model of change across time between control and *Arid1b* mutant mouse incisor mesenchymal cells using Monocle3 pseudotime analysis (Supplementary fig. 4a, b). This model confirmed the sequence of cell differentiation, showing that MSCs are the earliest (least differentiated) cells, followed by TACs and dental pulp cells, which are later and more differentiated cells. This data aligns with the in vivo dental mesenchymal cell differentiation axis. To gain further insights into the stem cell fate transition of dental mesenchymal cells, we conducted GeneSwitches analysis to identify the order of genes that are activated or repressed in the specific lineages[30]. We acquired sets of genes that switched on/off along the MSC to TAC trajectory in both control and *Arid1b* mutant samples (Supplementary Fig. 4c, d). This analysis helped us identify cluster-specific markers, such as *Igfbp6* for MSCs and *Pclaf* for TACs, which we compared between control and

*Arid1b* mutant samples (Supplementary Fig. 4e, f). Furthermore, we visualized and compared the patterns of common markers switching on/off along the MSC-TAC pseudotime trajectory between control and *Arid1b* mutant samples. Interestingly, the majority of switched-off genes displayed slightly earlier pseudo-timepoints in the *Arid1b* mutant, whereas the genes switched-on in TACs exhibited significantly earlier activation in the *Arid1b* mutant. This was especially true of proliferation-related genes, such as *Mki67*, *Top2a*, and *Cdk1* (Fig. 2m). This data indicated that loss of *Arid1b* may induce MSC proliferation. To confirm that the loss of *Arid1b* stimulates the proliferation of slow-cycling MSCs in vivo, we performed co-immunostaining of LRCs and Ki67 in the mouse incisor. The results revealed an increased number of LRC+ cells co-labeled with Ki67+ in *Arid1b* mutant mice compared to controls (Fig. 2n–r), indicating that the loss of *Arid1b* leads to the proliferation of MSCs. Taken together, these results suggested that the loss of *Arid1b* disturbs MSC quiescence and leads to the proliferation of MSCs.

## ARID1B directly suppresses BAF subunit *Bcl11b* expression in the MSC region to maintain tissue homeostasis

To investigate the downstream mechanisms of ARID1B in regulating the MSC population, we performed bulk RNA-seq to compare transcriptional profiles between control and *Gli1-CreER;Arid1b^{fl/fl}* mouse incisors. Hierarchical clustering and volcano plots revealed distinct transcriptional profiles between the two groups, identifying 155 upregulated and 55 downregulated genes (false discovery rate [FDR] ≤ 0.1; fold change < −1.5 or > 1.5) in *Gli1-CreER;Arid1b^{fl/fl}* mouse incisors (Fig. 3a, Supplementary Fig. 5a). We then conducted a comprehensive analysis and listed the top 20 differentially expressed genes and significantly changed signaling pathways based on the upregulated and downregulated genes (Fig. 3b, Supplementary Fig. 5b, c). Using the scRNA-seq data, we plotted the top 20 changed genes (Supplementary Fig. 5d). Among these genes, *Bcl11b* showed specific enrichment in the MSC and proximal dental follicle (PDF) clusters, colocalizing with *Gli1* expression (Fig. 3c). We compared its expression level within dental mesenchymal cells and observed its upregulation in *Arid1b* mutant mouse incisors (Fig. 3d). To confirm the expression pattern of *Bcl11b* in vivo, we colocalized *Bcl11b* with GLI1+ cells and found that *Bcl11b* is expressed in the proximal region of the mouse incisor surrounding the NVB and colocalized with GLI1+ cells (Fig. 3e, h). Furthermore, we confirmed the upregulation of *Bcl11b* expression in *Arid1b* mutant mouse incisors compared to the control (Fig. 3f, g, i–k). Taken together, these findings gave a strong indication that *Bcl11b* might act as a functional downstream target of ARID1B to regulate MSC quiescence. Additionally, considering that *Bcl11b* encodes a subunit of the BAF complex, it is essential to investigate the inter-regulation among the BAF subunits.

To investigate how ARID1B suppresses *Bcl11b* expression, we conducted single-cell transposase-accessible chromatin sequencing

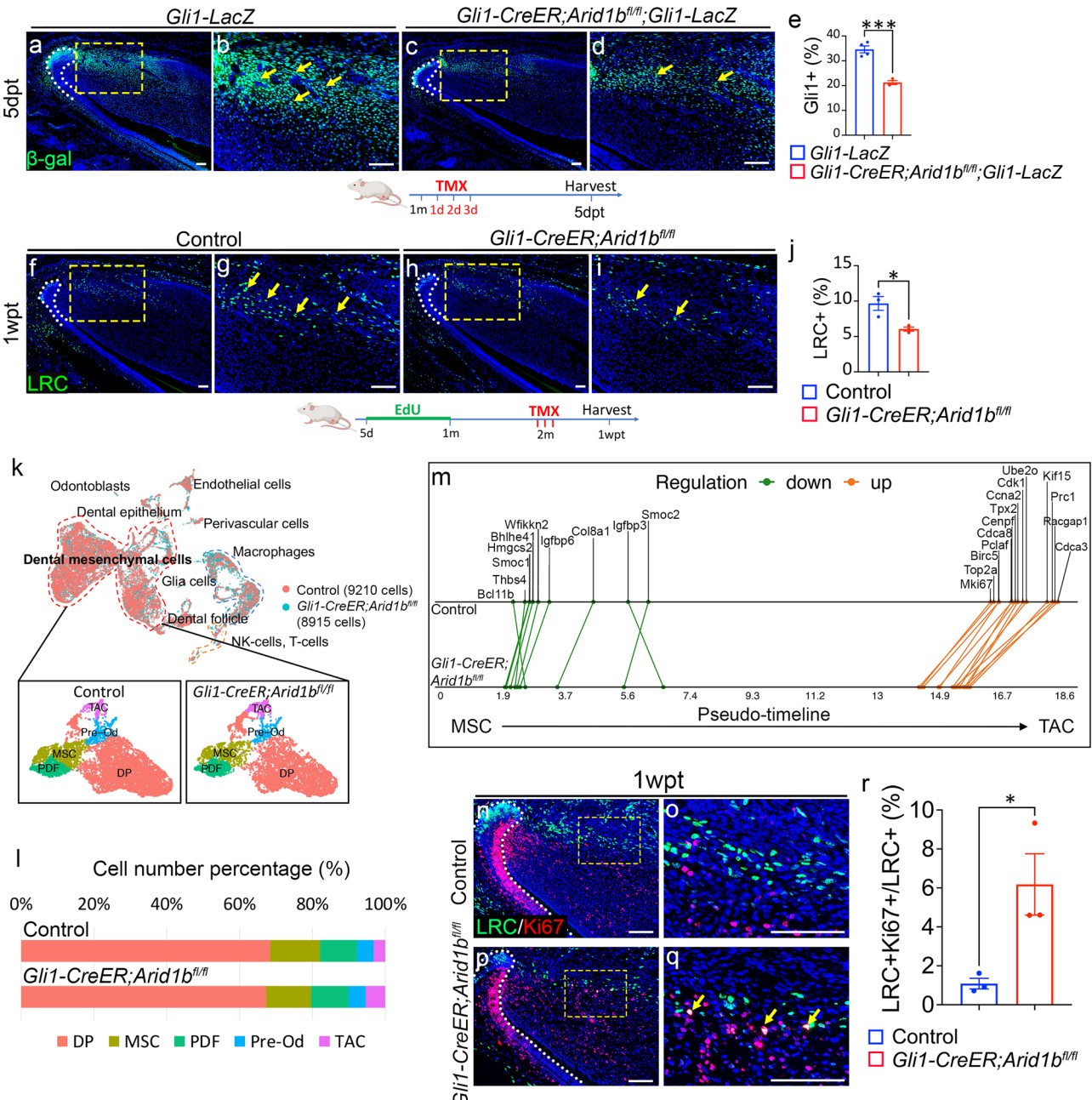

**Fig. 2 | Loss of *Arid1b* disturbs MSCs' quiescence and leads to their proliferation. a–d** Immunostaining of β-gal (green) in *Gli1-LacZ* and *Gli1-CreER;Arid1b^{fl/fl};Gli1-LacZ* mouse incisors. **b, d** represent the high-magnification images of the boxes in (**a, c**). White dotted lines outline the cervical loop. Yellow arrows point to the positive cells. dpt, day post-tamoxifen injection. **e** Quantification of GLI1+ cells in dental mesenchyme. Data are mean ± SEM, *Gli1-LacZ* n = 4, *Gli1-CreER;Arid1b^{fl/fl};Gli1-LacZ* n = 3, unpaired two-tailed Student's t-test. p = 0.0007. **f–i** EdU staining of the LRCs (green) of incisors from control (**f, g**) and *Gli1-CreER;Arid1b^{fl/fl}* (**h, i**). **g, i** represent the high-magnification images of the boxes in (**f, h**). White dotted lines outline the cervical loop. Yellow arrows point to the positive cells. Schematics of tamoxifen induction and EdU labeling protocol under (**a–d, f–i**), created with BioRender.com released under a Creative Commons Attribution-NonCommercial-NoDerivs 4.0 Internatioanl license. **j** Quantification of LRCs in dental mesenchyme. Data are mean ± SEM, *n* = 3, unpaired two-tailed Student's t-test. p = 0.0238. **k** UMAP visualization of integrated scRNA-seq and the subclusters in dental mesenchymal cells from control and *Gli1-CreER;Arid1b^{fl/fl}* mouse incisors. MSC,

mesenchymal stem cell; PDF, proximal dental follicle; TAC, transit-amplifying cell; DP, dental pulp; Pre-Od, pre-odontoblast. **l** Cell number percentage of dental mesenchymal cell clusters in control and *Gli1-CreER;Arid1b^{fl/fl}* samples based on scRNA-seq data. **m** GeneSwitches comparison of pseudotime along the MSC to TAC trajectory between control and *Gli1-CreER;Arid1b^{fl/fl}* mouse incisor samples. Green dots indicate the pseudo-timepoints of genes that have been switched off, while orange dots indicate the genes that have been switched on. The green and orange lines connect the same genes. **n–q** Co-immunostaining of LRCs (green) and Ki67 (red) in control (**n, o**) and *Gli1-CreER;Arid1b^{fl/fl}* (**p, q**) mouse incisors. **o, q** represent high-magnification images of boxes in (**n, p**). White dotted lines outline the cervical loop. Yellow arrows point to the co-stained positive cells. **r** Quantification of the percentage of LRC+/Ki67+ cell number in LRC+ cells from control and *Gli1-CreER;Arid1b^{fl/fl}* mouse incisors. Data are mean ± SEM, *n* = 3, unpaired two-tailed Student's t-test. p = 0.0333. Source data are provided as a Source Data file. Scale bars: 100 μm.

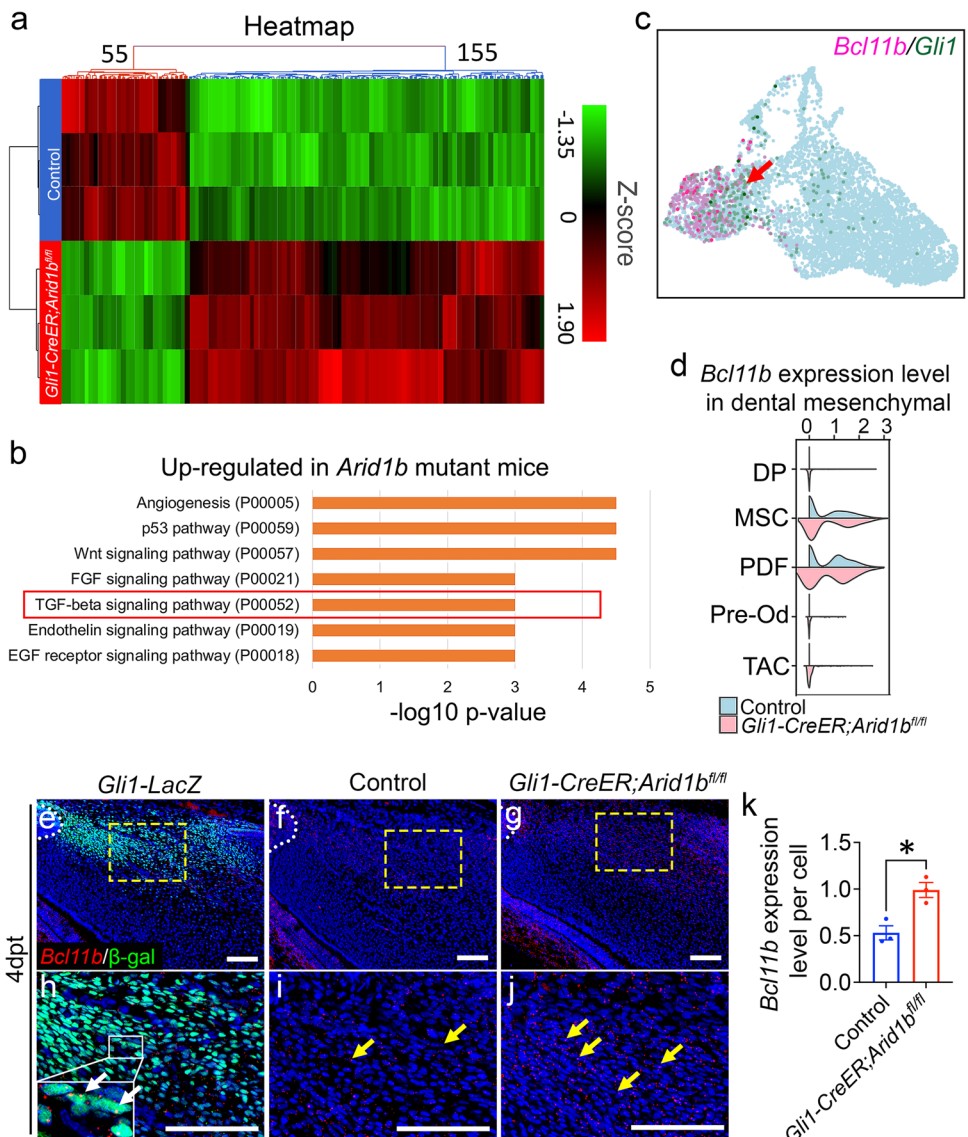

**Fig. 3 | Upregulation of *Bcl11b* following the loss of *Arid1b* in mouse incisors.**
**a** Heatmap of bulk RNA-seq data for the proximal region of control and *Gli1-CreER;Arid1b^fl/fl* mouse incisor at 4 days post-tamoxifen induction. **b** Top 7 signaling pathways identified by GO analysis using upregulated genes identified from bulk RNA-seq analysis. **c** FeaturePlot of *Bcl11b* and *Gli1* expression in the control mouse incisor sample. The red arrow points to the co-localized positive cells. **d** ViolinPlot of *Bcl11b* expression levels in control and *Gli1-CreER;Arid1b^fl/fl* mouse incisor mesenchymal cells. **e, h** In situ hybridization of *Bcl11b* (red) and immunostaining of β-gal (green) in *Gli1-LacZ* mouse incisor. **h** represents a high-magnification image of

the box in (**e**). The white dotted line outlines the cervical loop. White arrows point to the positive cells. *n* = 3. **f, g, i, j** Comparison of *Bcl11b* (red) expression in control (**f, i**) and *Gli1-CreER;Arid1b^fl/fl* (**g, j**) mouse incisors at 4 days post-tamoxifen induction. **i, j** represent high-magnification images of the boxes in (**f, g**), respectively. The white dotted line outlines the cervical loop. Yellow arrows point to the positive signals. **k** Quantification of the *Bcl11b* expression level per cell in (**i, j**). Data are mean ± SEM, *n* = 3, unpaired two-tailed Student's t-test. *p* = 0.0142. Source data are provided as a Source Data file. Scale bars, 100 μm.

(scATAC-seq) using the proximal region of incisors from both control and *Gli1-CreER;Arid1b^fl/fl* mice 4 days after tamoxifen induction. By analyzing and integrating the sequencing data from control and *Arid1b* mutant mice using Signac and marker analysis, we identified distinct cell clusters (Fig. 4a, Supplementary Fig. 6). *Gli1* putative gene activity was found to be significantly enriched in the MSC population (Fig. 4b). To compare the differences in chromatin accessibility specifically within the proximal mesenchyme cells between the control and *Arid1b* mutant mice, we categorized the cells into Control-MSC, Mutant-MSC, and other clusters (Fig. 4c). We generated the normalized signals to visualize the DNA accessibility and annotated peaks for *Bcl11b*. The scATAC-seq results revealed that the Mutant-MSC cluster had higher peaks compared to the Control-MSC cluster at the promoter region of *Bcl11b*, indicating increased chromatin accessibility at the promoter

region following the loss of *Arid1b* (Fig. 4d). These results provided evidence that ARID1B functions as a suppressor of *Bcl11b* gene expression in the mouse incisor.

To investigate whether ARID1B directly binds to the *cis*-regulatory elements of *Bcl11b* to regulate its expression, we compared the peaks between control and *Arid1b* mutant scATAC-seq data, and identified a reduced peak call at Chr12: 107964756-107965631, corresponding to the *Bcl11b* third intron region in *Arid1b* mutant sample (Fig. 4d). To identify whether ARID1B can bind to this 865 base-pair (bp) region, we designed primers within this region and performed ARID1B chromatin immunoprecipitation followed by quantitative PCR (CHIP-qPCR) (Fig. 4e). The CHIP-qPCR results revealed significantly higher DNA binding of ARID1B CHIP compared to the IgG control in the +101 bp to +262 bp binding region (as shown in Fig. 4e between the black arrows),

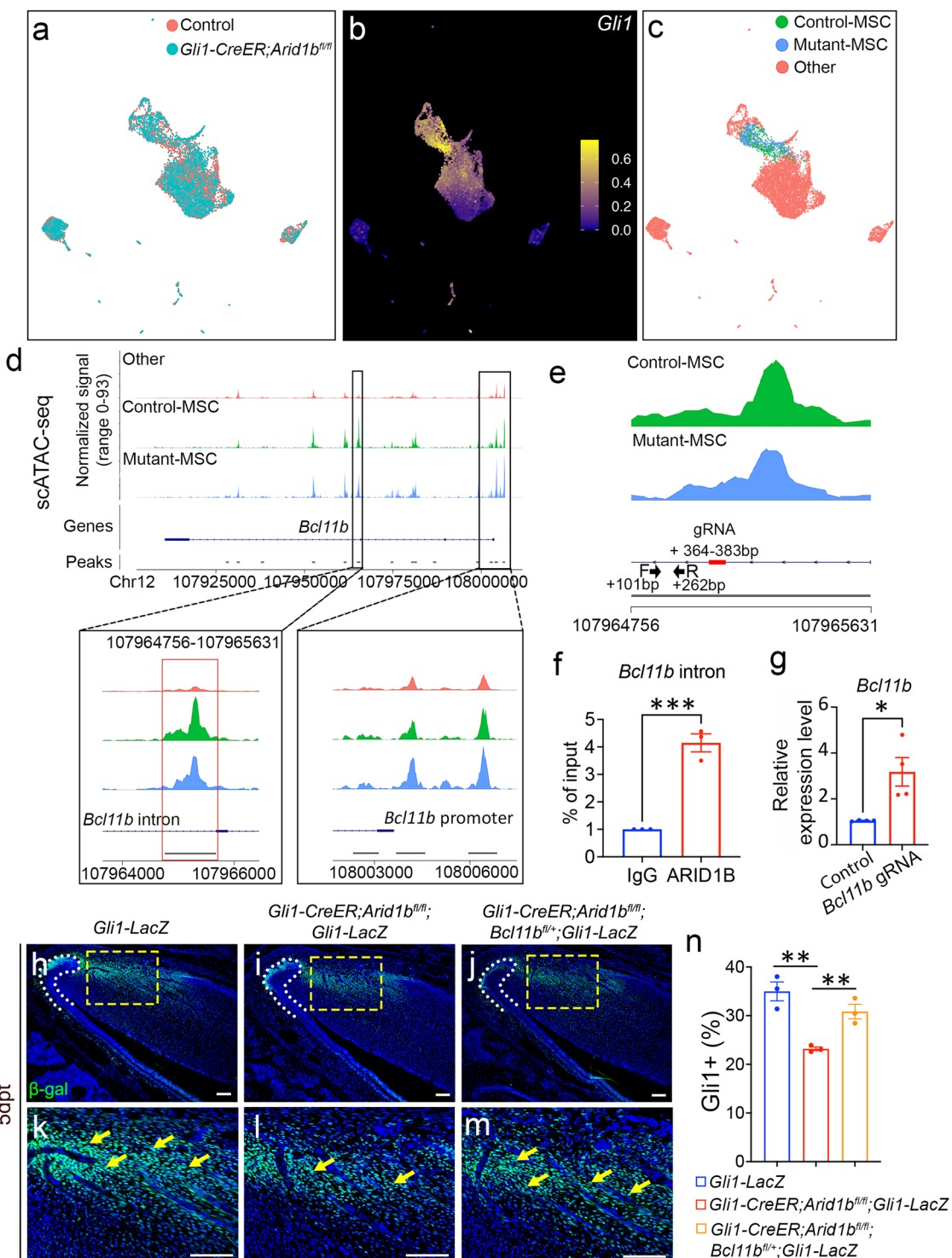

**Fig. 4 | ARID1B directly suppresses *Bcl11b* expression through binding to its intron region. a** UMAP visualization of integrated scATAC-seq data from control and *Gli1-CreER;Arid1b^{fl/fl}* mouse incisors at 4 days post-tamoxifen induction. **b** FeaturePlot of *Gli1* putative gene activity of scATAC-seq data. **c** Re-clustered UMAP visualization with control-MSC, mutant-MSC, and other clusters. **d** Peak calling from scATAC-seq at *Bcl11b* gene. The annotated open chromatin of the changed intron region and promoter region of *Bcl11b* are enlarged in the box below. **e** CHIP-qPCR primers and CRISPRi gRNA design at the targeted intron region of *Bcl11b*. Black arrows indicate the position and direction of forward and reverse primers. Red bolded line indicates the gRNA position. **f** CHIP assay with ARID1B antibody (or immunoglobulin G [IgG]), followed by qPCR with designed primers at the intron region of *Bcl11b*. Data are mean ± SEM, *n* = 3, unpaired two-tailed Student's t-test. *p* = 0.0007. **g** RT-qPCR analysis of *Bcl11b* expression following CRISPRi

treatment with vectors (generated by Origene Technologies) containing control and *Bcl11b* gRNA. Data are mean ± SEM, *n* = 4, unpaired two-tailed Student's t-test. *p* = 0.0141. **h–m** β-gal (green) immunostaining of incisors from *Gli1-LacZ* (**h, k**), *Gli1-CreER;Arid1b^{fl/fl};Gli1-LacZ* (**i, l**), and *Gli1-CreER;Arid1b^{fl/fl};Bcl11b^{fl/+};Gli1-LacZ* (**j, m**) mice 5 days post-tamoxifen induction. **k, l, m** represent high-magnification images of boxes in (**h–j**), respectively. White dotted lines outline the cervical loop. Yellow arrows point to the positive cells. Scale bars, 100 μm. (**n**) Quantification of the GLI1+ cells in dental mesenchyme of incisors from *Gli1-LacZ*, *Gli1-CreER;Arid1b^{fl/fl};Gli1-LacZ*, and *Gli1-CreER;Arid1b^{fl/fl};Bcl11b^{fl/+};Gli1-LacZ* mice. Data are mean ± SEM, *n* = 3, unpaired two-tailed Student's t-test. *Gli1-LacZ* vs *Gli1-CreER;Arid1b^{fl/fl};Gli1-LacZ*, *p* = 0.0039; *Gli1-CreER;Arid1b^{fl/fl};Gli1-LacZ* vs *Gli1-CreER;Arid1b^{fl/fl};Bcl11b^{fl/+};Gli1-LacZ*, *p* = 0.0078. Source data are provided as a Source Data file.

suggesting that ARID1B can directly bind to the intron of *Bcl11b* to suppress its expression (Fig. 4f). To further validate the functional relevance of this binding site, we performed CRISPRi to specifically target the binding region and assess its effect on the transcriptional activity. It is known that the dCas9 for CRISPRi exerts its binding activity within its gRNA target position with a range of ±150 bp[31,32]. Accordingly, we designed a gRNA around the qPCR-targeted region with high targeting efficiency and low off-target rate (Fig. 4e). Following CRISPRi treatment of primary cells, we observed a significant increase in the *Bcl11b* expression level in the group treated with the vector containing *Bcl11b* gRNA when compared to the group treated with the control vector (Fig. 4g), indicating that the expression level of *Bcl11b* was upregulated when there was interference with the ARID1B binding site.

Furthermore, to confirm the role of *Bcl11b* as a key downstream mediator of ARID1B in regulating MSC homeostasis, we generated *Gli1-CreER;Arid1b^{fl/fl};Bcl11b^{fl/+};Gli1-LacZ* mice. By comparing the numbers of GLI1+ MSCs with *Gli1-LacZ* and *Gli1-CreER;Arid1b^{fl/fl};Gli1-LacZ* mice, we aimed to assess the impact of *Bcl11b* expression on MSC populations in vivo. Through immunostaining, we observed that the reduction of *Bcl11b* expression in *Gli1-CreER;Arid1b^{fl/fl};Bcl11b^{fl/+}* mice led to the restoration of GLI1+ cells (Fig. 4h–n). These compelling findings strongly supported the notion that *Bcl11b* serves as the functional downstream target of ARID1B, playing a crucial role in regulating MSC dynamics. Moreover, our data unveiled the direct inter-regulatory interactions between ARID1B and BCL11B during their functional regulation.

## BCL11B directly regulates Activin A subunit gene *Inhba* to modulate Activin signaling

Using the differentially expressed genes identified from bulk RNA-seq, we investigated the significant changes in signaling pathways following the loss of *Arid1b* in the mouse incisor. Through gene ontology (GO) analysis, we identified highly enriched upregulated and downregulated signaling pathways based on the genes that showed significant increased or decreased expression in *Arid1b* mutant mice, respectively (Fig. 3b, Supplementary Fig. 5c). Among these pathways, we observed significant changes in angiogenesis and its related p53 pathway, as well as FGF, Wnt, Endothelin, and EGF signaling, with both upregulated and downregulated genes. This suggested that these signaling pathways may not be directly regulated by the loss of *Arid1b*. TGF-β signaling exhibited significant upregulation in *Arid1b* mutant mice, indicating that this could be a downstream signaling pathway that undergoes substantial modulation following the loss of *Arid1b*. We therefore explored the role of TGF-β signaling and sought to determine the specific ligands and receptors involved in maintaining MSC homeostasis.

Previous studies on *Bcl11b*-deficient mice have shown downregulation of TGF-β family ligands during embryonic incisor development, including BMP4 and activin, leading to disruption of TGF-β signaling[33]. We thus proposed that TGF-β signaling superfamily may exert as a downstream regulatory effect of BCL11B. To test our hypothesis, we compared the expression patterns and levels of ligands and receptors within the TGF-β superfamily, including TGF-β signaling, Activin/Inhibin signaling, and BMP signaling, using integrated scRNA-seq and bulk RNA-seq data. Our analysis did not reveal significant changes in BMP signaling (Supplementary Fig. 7). However, we observed significant upregulation of *Tgfbr1* and *Inhba* following the loss of *Arid1b* in the mouse incisor from the bulk RNA-seq data. *Inhba* encodes a subunit for the Activin signaling ligand, activin A. Notably, when we plotted these ligands and receptors in the scRNA-seq data, we observed that *Inhba* was predominantly enriched in the PMC and dental follicle clusters, which coincided with *Bcl11b* expression (Fig. 5a, Supplementary Fig. 8a, b). However, *Tgfbr1* exhibited widespread expression in the dental mesenchyme, and according to the JASPAR

transcription factor (TF) binding sites prediction tool integrated into the UCSC Genome Brower, there is no predicted binding site for BCL11B at the promoter region of *Tgfbr1* (Supplementary Fig. 8b, c). This indicated that the increase of *Tgfbr1* may not be directly regulated by BCL11B. Thus, we turned our focus to *Inhba* as a potential target of BCL11B within the TGF-β superfamily. To further investigate the expression of *Inhba* and its colocalization with *Bcl11b* in vivo, we performed in situ hybridization. The staining revealed the colocalization of *Inhba* and *Bcl11b* within the MSC region and the dental follicle of the control incisors (Fig. 5b–d), indicating BCL11B may directly regulate *Inhba* expression.

To elucidate the mechanism by which BCL11B, acting as a transcription factor, mediates the regulation of *Inhba* expression, we investigated the direct binding of BCL11B to the promoter region of *Inhba*. Utilizing the JASPAR TF binding sites prediction, we identified two potential binding sites located within 1.5 kb upstream of the transcription start site (TSS) in the promoter region (Fig. 5e). We specifically targeted the binding site closer to the TSS, as it exhibited a higher predicted binding score according to JASPAR prediction. We first compared the scATAC-seq data for *Inhba* and observed higher peaks in the Mutant-MSC cluster compared to the Control-MSC cluster at the promoter region of *Inhba* (Fig. 5f). This data indicated increased chromatin accessibility at the *Inhba* promoter region in the *Arid1b* mutant mice. Furthermore, we designed primers according to the predicted binding motif and performed BCL11B ChIP-qPCR in control mouse incisors. The result revealed significantly higher DNA binding of BCL11B CHIP than the IgG control at the promoter region of *Inhba*, indicating that BCL11B directly binds to the promoter region and regulates *Inhba* gene expression (Fig. 5g). These findings provided evidence supporting the direct regulation of *Inhba* expression by BCL11B.

Furthermore, we examined the ligand activin A level through immunostaining and found that it was primarily deposited in the TAC region, odontoblasts, and dental pulp cells near the odontoblasts in control incisors. We noticed an ectopic upregulation of activin A in the MSC region following the loss of *Arid1b*, which was restored to normal after reducing *Bcl11b* level in *Arid1b* mutant mice (Fig. 5h–m). These findings indicated that BCL11B functionally and directly regulates the expression of the activin A subunit *Inhba*. Activin A, as the downstream target of BCL11B, undergoes ectopic upregulation in the MSC region within the *Gli1-CreER;Arid1b^{fl/fl}* mouse incisor. To further explore the impact of ACTIVIN A upregulation in *Arid1b* mutant mice and its potential effect in the reduction of GLI1+ MSCs, we conducted an explant culture experiment. Using IgG or neutralizing ACTIVIN A antibody-loaded beads, we treated *Gli1-LacZ* and *Gli1-CreER;Arid1b^{fl/fl};Gli1-LacZ* mouse incisor explants (Fig. 5n). Immunostaining confirmed the decrease of GLI1+ MSCs in the *Arid1b* mutant mouse incisors compared to controls treated with IgG (Fig. 5o, p, r). Notably, treatment of *Gli1-CreER;Arid1b^{fl/fl};Gli1-LacZ* mouse incisor explants with the ACTIVIN A antibody restored the normal level of GLI1+ MSCs, in comparison to IgG treatment (Fig. 5p–r). These findings strongly suggested that ACTIVIN A serves as a functional downstream factor capable of maintaining MSC homeostasis in mouse incisors affected by the loss of *Arid1b*.

## Loss of *Arid1b* ectopically activates non-canonical Activin signaling, p-ERK, impairing MSC homeostasis

Activin A, the ligand that activates the Activin signaling pathway, belongs to the TGF-β superfamily. Activin A binds to the extracellular domain of a type II receptor, ActRIIA or ActRIIB, then forms a complex with the type I receptor, ActRI or TGFBR1. The type I receptor phosphorylates downstream signaling molecules, leading to the activation of the Activin signaling pathway. Upon ligand binding, TGF-β/Activin receptors activate intracellular signaling pathways such as the

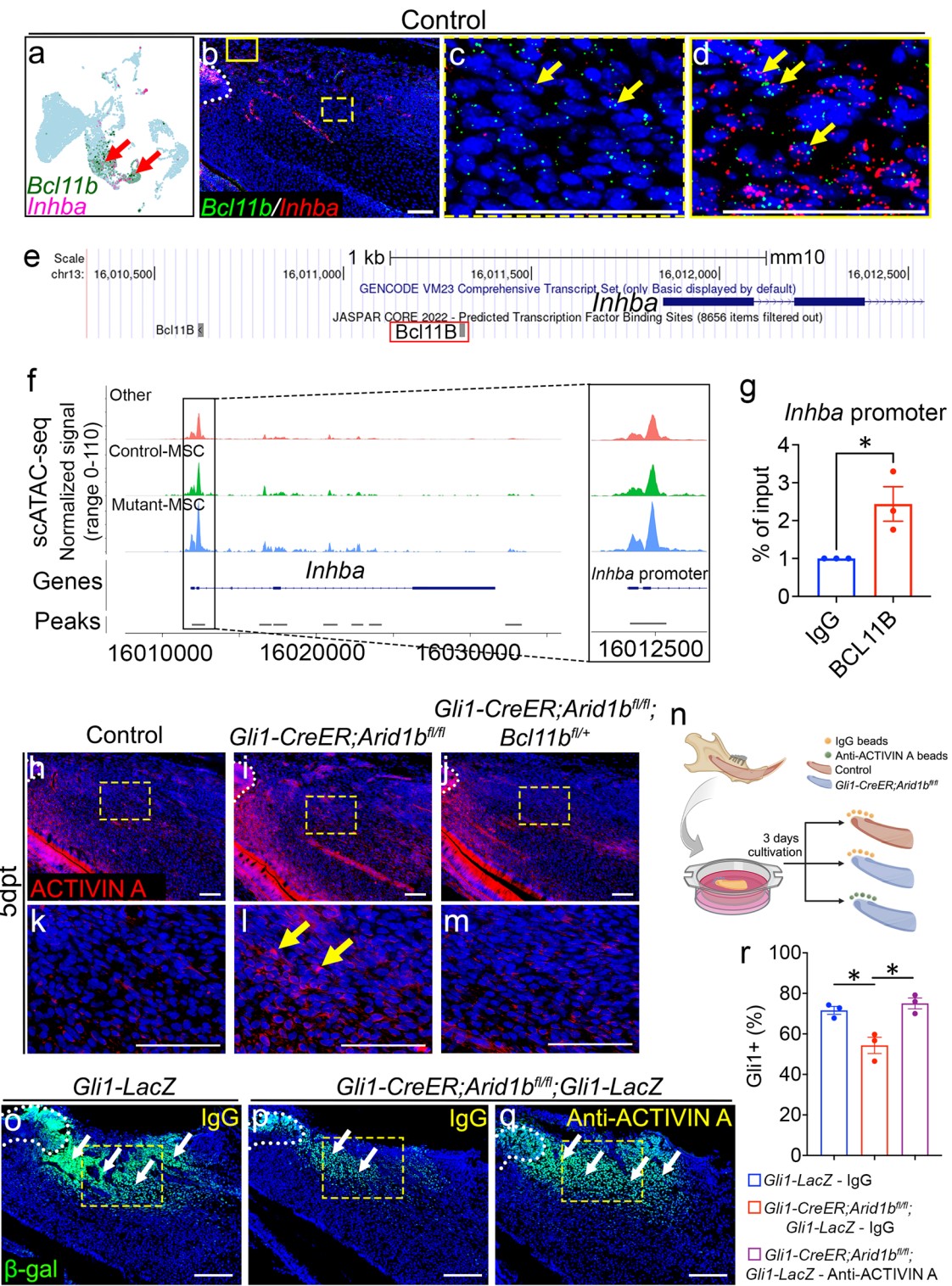

canonical Smad-dependent pathway via Smad2/Smad3, and non-Smad pathways including p38 MAPK, JNK, ERK, AKT, NF-κB, and COFILIN signaling[34,35]. These signaling pathways act in a context-dependent manner, leading to different cellular responses. To investigate changes in Activin signaling activity in the *Gli1-CreER;Arid1b^{fl/fl}* mouse incisor compared to the control, we initially conducted western blot to assess the changes in the signaling pathways. Interestingly, we observed that p-JNK, p-ERK, and p-COFILIN signaling were elevated following the loss of *Arid1b*, while p-SMAD2, the readout of canonical TGF-β/Activin

signaling, exhibited no significant changes (Fig. 6a, Supplementary Fig. 9). To further corroborate these findings and explore where these signaling pathways were activated in vivo, we performed immunostaining for p-JNK, p-ERK, and p-COFILIN in control and *Arid1b* mouse incisors. Our data revealed that p-COFILIN and p-JNK were specifically activated within the MSC region, as evidenced by their co-localization with GLI1+ cells, and exhibited heightened expression upon *Arid1b* loss (Fig. 6b−m). Interestingly, p-ERK, which was not activated in the MSC region of the control incisor, exhibited ectopic activation in the MSC

**Fig. 5 | BCL11B directly regulates the expression of *Inhba*, a subunit of Activin A, which serves as a ligand of the Activin signaling pathway. a** FeaturePlot shows the colocalization of *Bcl11b* and *Inhba* in the control scRNA-seq data. Red arrows point to the co-localized positive cells. **b−d** in situ hybridization of *Bcl11b* (green) and *Inhba* (red) in the proximal region of the mouse incisor. **c** and **d** represent high-magnification images of boxes in (**b**). White dotted line outlines the cervical loop. Yellow arrows point to the co-stained positive cells. *n* = 3. **e** UCSC binding prediction of BCL11B binding motif to the promoter region of *Inhba*. The red box emphasizes the binding site with the highest score. **f** Peak calling from scATAC-seq at *Inhba*. The annotated open chromatin at the promoter region of *Inhba* are enlarged at the right-side box. **g** CHIP assay with BCL11B antibody (or immunoglobulin G [IgG]), followed by qPCR with primers designed for the promoter region of *Inhba*. Data are mean ± SEM, *n* = 3, unpaired two-tailed Student's t-test. *p* = 0.0337. **h−m** Immunostaining of Activin A (red) in the incisors from control (**h, k**), *Gli1-CreER;Arid1b^{fl/fl}* (**i, l**), and *Gli1-CreER;Arid1b^{fl/fl};Blc11b^{fl/+}*

(**j, m**) mice 5 days post-tamoxifen induction. **k, l, m** represent high-magnification images of boxes in (**h−j**), respectively. White dotted lines outline the cervical loop. Yellow arrows indicate positive cells. *n* = 3. **n** Schematic drawing of the proximal ends of mouse incisors in explant culture created with BioRender.com released under a Creative Commons Attribution-NonCommercial-NoDerivs 4.0 Internatioanl license. **o−q** β-gal (green) immunostaining of incisor explants of *Gli1-lacZ* treated with IgG beads (**o**), *Gli1-CreER;Arid1b^{fl/fl};Gli1-lacZ* treated with IgG beads (**p**), and *Gli1-CreER;Arid1b^{fl/fl};Gli1-lacZ* treated with anti-ACTIVIN A beads (**q**). White dotted lines outline the cervical loop. White arrows indicate positive cells. Yellow boxes indicate the area for quantification. *n* = 3. **r** Quantification of the GLI1+ cells in dental mesenchyme of incisor explants. Data are mean ± SEM, *n* = 3, unpaired two-tailed Student's t-test. *Gli1-LacZ* - IgG vs *Gli1-CreER;Arid1b^{fl/fl}; Gli1-LacZ* - IgG, *p* = 0.0176; *Gli1-CreER;Arid1b^{fl/fl};Gli1-LacZ* - IgG vs *Gli1-CreER; Arid1b^{fl/fl};Gli1-LacZ* - ACTIVIN A, *p* = 0.0125. Source data are provided as a Source Data file. Scale bars, 100 μm (**b, h−m, o−q**); 200 μm (**c, d**).

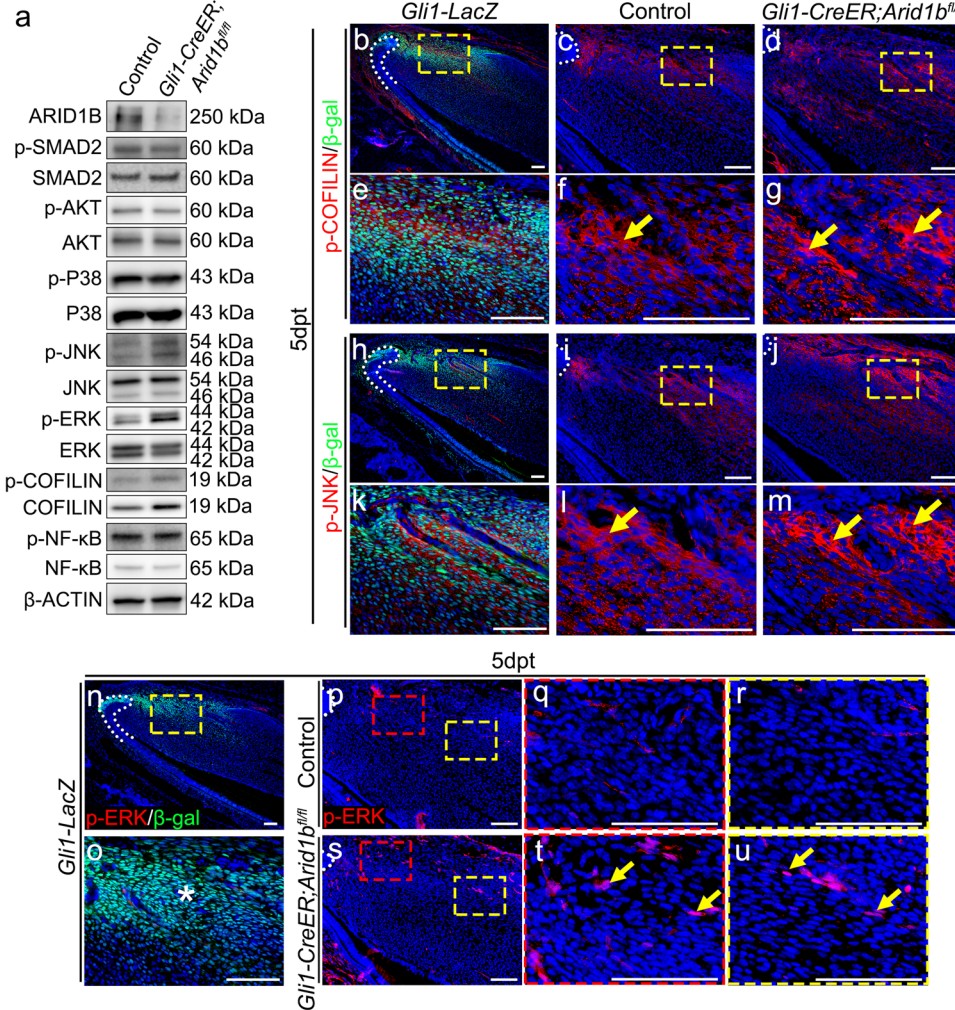

**Fig. 6 | Non-canonical Activin signaling pathways are elevated after loss of *Arid1b*. a** Western blot of ARID1B, p-SMAD2, SMAD2, p-AKT, AKT, p-P38, P38, p-ERK, ERK, p-JNK, JNK, p-COFILIN, COFILIN, p-NF-κB, NF-κB, and β-ACTIN in proximal incisor mesenchyme from control and *Gli1-CreER;Arid1b^{fl/fl}* mice. **b−g** Immunostaining of p-COFILIN (red) with β-gal (green) at *Gli1-LacZ* mouse incisor (**b, e**) and comparison of p-COFILIN expression in control (**c, f**) and *Gli1-CreER;Arid1b^{fl/fl}* (**d, g**) mouse incisors. **e−g** represent high-magnification images of boxes in (**b−d**), respectively. White dotted lines outline the cervical loop. Yellow arrows indicate positive cells. *n* = 3. **h−m** Immunostaining of p-JNK (red) with β-gal (green) in *Gli1-LacZ* mouse incisor (**h, k**) and comparison of p-JNK expression in control (**i, l**) and *Gli1-CreER;Arid1b^{fl/fl}*

(**j, m**) mouse incisors. **k−m** represent high-magnification images of boxes in (**h−j**), respectively. White dotted lines outline the cervical loop. Yellow arrows indicate positive cells. *n* = 3. **n−o** Immunostaining of p-ERK (red) with β-gal (green) in *Gli1-LacZ*. **o** represents the high-magnification image of (**n**). White dotted line outlines the cervical loop. The asterisk in (**o**) represents the absence of co-stained cells. **p−u** Immunostaining of p-ERK (red) in control (**p−r**) and *Gli1-CreER;Arid1b^{fl/fl}* (**s−u**) mouse incisors. **q, r** represent the high-magnification image of boxes in (**p**), **t, u** represent the high-magnification image of boxes in (**s**). White dotted lines outline the cervical loop. Yellow arrows indicate positive cells. *n* = 3. Scale bars, 100 μm.

region after loss of *Arid1b* (Fig. 6n–u). Our data provided robust evidence suggesting that non-canonical Activin signaling pathways play a pivotal role in the regulation of MSC homeostasis.

Additionally, we investigated where p-SMAD2 was activated in both control and *Gli1-CreER;Arid1b^fl/fl* mouse incisors. Our results demonstrated that p-SMAD2 was primarily activated in dental pulp cells and adjacent to GLI1+ cells (Supplementary Fig. 10a–h). This finding identified that Smad-dependent TGF-β/Activin signaling is confined within the differentiated dental pulp compartment, delineating the boundary between the MSCs and dental pulp cells in the mouse incisor. This spatial segregation remained unaltered following the loss of *Arid1b* at 5 days post-tamoxifen induction.

To determine whether the elevated non-canonical Activin signaling was the cause of the incisor defect observed in the *Gli1-CreER;Arid1b^fl/fl* mice, we generated *Gli1-CreER;Arid1b^fl/fl;Tgfbr1^fl/+* mice as a rescue model. This approach was based on the significant upregulation of *Tgfbr1* following the loss of *Arid1b* (Supplementary Fig. 8b), and the notion that TGFBR1 plays a crucial role in TGF-β/Activin signaling by forming receptor dimers with TGFBR2 and ActRII, transporting signals from intercellular to intracellular. Through in situ hybridization, we detected *Tgfbr1* signal widely expressed in the dental mesenchyme, which was upregulated in *Arid1b* mutant mouse incisors compared to the control (Supplementary Fig. 10i–n). Tamoxifen was injected at 1 month of age to induce Cre activity. At 3 months post-tamoxifen induction, microCT images revealed that the narrowed dental pulp cavity observed in *Gli1-CreER;Arid1b^fl/fl* mice were completely rescued in *Gli1-CreER;Arid1b^fl/fl;Tgfbr1^fl/+* mouse incisors. Moreover, HE staining further confirmed that the incisor defects in the *Arid1b* mutant mice, such as stacked dentin at the cervical loop, misaligned odontoblasts, and thicker dentin, were also rescued in *Gli1-CreER;Arid1b^fl/fl;Tgfbr1^fl/+* mice. The odontoblast marker *Dspp* showed that the premature differentiation of odontoblasts was eliminated in the rescue mouse incisor as well (Fig. 7a–i). We assessed the numbers of GLI1+ cells in *Gli1-LacZ*, *Gli1-CreER;Arid1b^fl/fl;Gli1-LacZ* and *Gli1-CreER;Arid1b^fl/fl;Tgfbr1^fl/+;Gli1-LacZ* mice 5 days after tamoxifen induction and confirmed that the reduced number of GLI1+ MSCs observed in the *Gli1-CreER;Arid1b^fl/fl;Gli1-LacZ* mouse incisors was restored in the *Gli1-CreER;Arid1b^fl/fl;Tgfbr1^fl/+;Gli1-LacZ* mice (Fig. 7j–p).

Furthermore, to investigate whether the dysregulated non-canonical Activin signaling had been restored in the *Tgfbr1* rescue mouse model, we performed western blot analysis of p-JNK, p-ERK, and p-COFILIN on control, *Gli1-CreER;Arid1b^fl/fl*, and *Gli1-CreER;Arid1b^fl/fl;Tgfbr1^fl/+* mouse incisors. Notably, only the p-ERK signaling showed obvious restoration, suggesting that it plays a key role in the non-canonical Activin signaling associated with the loss of *Arid1b* in the mouse incisors (Fig. 7q). To validate that the p-ERK signaling pathway is the functionally downstream, we generated *Gli1-CreER;Arid1b^fl/fl;Erk2^fl/+;Gli1-LacZ* mice. By comparing GLI1+ cell populations among *Gli1-LacZ*, *Gli1-CreER;Arid1b^fl/fl;Gli1-LacZ*, and *Gli1-CreER;Arid1b^fl/fl;Erk2^fl/+;Gli1-LacZ* mouse incisors, we observed the restoration of GLI1+ cells in the *Erk2* rescue mouse model (Fig. 7r–x). Ultimately, the loss of *Arid1b* led to the activation of p-ERK signaling in MSCs following activin A ligand binding to TGFBR1-associated receptors. The aberrant p-ERK signaling disrupted MSC homeostasis and diminished their population.

## Discussion

This study unveils the functional role of ARID1B in stem cells and provides novel insights into the regulatory network of ARID1B as a chromatin remodeler in governing MSC quiescence and proliferation, thereby contributing to proper tissue homeostasis. Our findings demonstrate that ARID1B functions as a repressor regulating the BAF complex subunit gene *Bcl11b* in the MSC region to maintain tissue homeostasis. Loss of *Arid1b* results in the upregulation of *Bcl11b*, leading to the direct induction of activin A and ectopic activation of

non-canonical Activin signaling via binding to TGFBR1-associated receptors, specifically through the p-ERK pathway. Overall, the ectopic activation of p-ERK signaling within MSCs after the loss of *Arid1b* impairs MSC quiescence and proliferation, ultimately impacting tissue homeostasis (Fig. 8). These findings shed light on the intricate molecular mechanisms underlying the regulatory role of ARID1B and its cellular function in MSC homeostasis.

The BAF complex plays a critical role in regulating various aspects of cellular processes, most notably chromatin remodeling, gene expression, and cell differentiation. Its involvement in controlling the fate and function of stem cells highlights its essential role in development, tissue homeostasis, and cellular plasticity. Notably, different subunits within the BAF complex exhibit distinct regulatory functions. Previous research has demonstrated that ARID1A, which is mutually exclusive from ARID1B in the BAF complex, regulates the cell cycle exit and differentiation of TACs to maintain tissue homeostasis in the mouse incisor[36]. Our investigations have revealed a distinct role for ARID1B in regulating MSC quiescence, suggesting that ARID1A and ARID1B perform different functions in mediating the reciprocal interaction between MSCs and TACs. Given the mutual exclusivity of ARID1A and ARID1B, we assessed the expression level of ARID1A in *Arid1b* mutant mouse incisors. Our analysis revealed expanded expression of ARID1A in MSCs and proximal dental follicle cells compared to the control (Supplementary Fig. 11a–d). To further elucidate the role of ARID1A in *Arid1b* mutant mice, we generated compound mutant mice with ARID1A haploinsufficiency and loss of *Arid1b* (*Gli1-CreER;Arid1b^fl/fl;Arid1a^fl/+*) and collected incisor samples 3 months after tamoxifen injection. HE staining of these samples revealed a more severe impairment of tissue homeostasis in the *Gli1-CreER;Arid1b^fl/fl;Arid1a^fl/+* mouse incisors, characterized by stacked dentin and disorganized odontoblasts (Supplementary Fig. 11e–j). These findings strongly imply that ARID1A may partially compensate for the loss of ARID1B function.

Moreover, researchers have utilized induced pluripotent stem cells (iPSCs) derived from Coffin-Siris patients with ARID1B haploinsufficiency and revealed that the ARID1B-BAF complex acts as a suppressor of numerous enhancers and genes within the NANOG and SOX2 networks, providing critical insights into the regulatory mechanisms underlying this transition[17]. In our study, ARID1B maintains MSC homeostasis primarily through the BCL11B-non-canonical Activin pathway. Our bulk RNA-seq analysis has revealed that three quarters of the genes are upregulated and only one quarter is downregulated following the loss of *Arid1b* in the mouse incisor, suggesting that ARID1B acts primarily as a suppressor in the regulation of its downstream genes. The gene regulatory network shown here provides compelling evidence for the identified role and regulatory mechanisms of ARID1B in MSC regulation. These insights not only contribute to our understanding of the broader functions and regulatory mechanisms of ARID1B but also serve as a valuable information for future studies investigating the role of ARID1B in other stem cell types and tissues.

Our study has revealed how ARID1B regulates *Bcl11b* expression, highlighting its pivotal role in maintaining MSC quiescence. The regulatory network involving the ARID1B-BCL11B-*Inhba* axis operates within the MSC region of the mouse incisor, contributing to the functional maintenance of tissue homeostasis. Notably, BCL11B has been extensively studied as an important transcription factor involved in regulating the immune system, including T cell development and identity, as well as the development of the skin, adipocytes, craniofacial region, and central nervous system[37–42]. However, understanding the role of BCL11B in regulating stem cell fate remains relatively limited. Our study has revealed that p-ERK is a downstream signaling pathway regulated by BCL11B. The ectopic activation of p-ERK in the MSC region leads to a decrease in the MSC population and disrupts tissue homeostasis. Importantly, our findings provide further insight

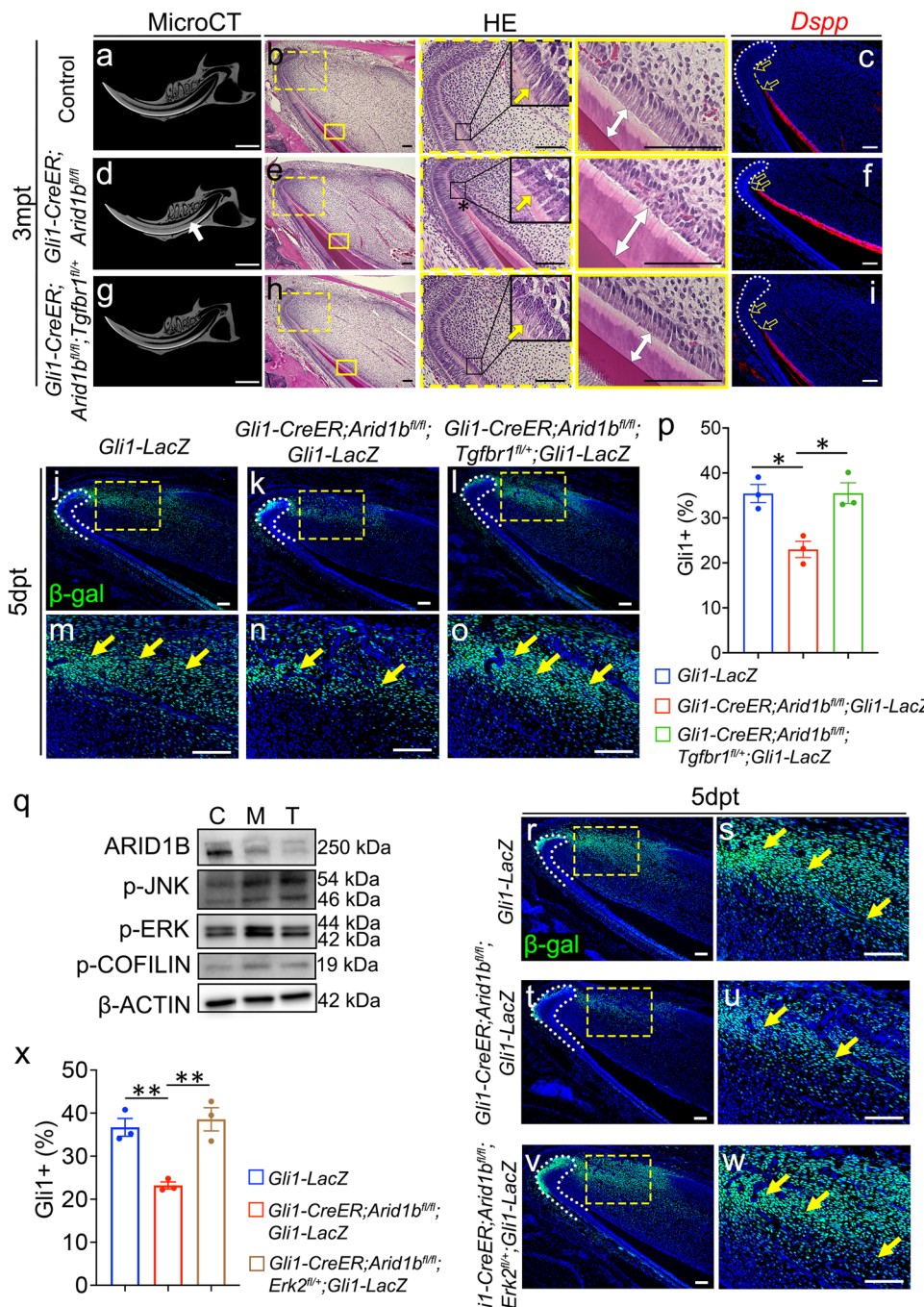

**Fig. 7 | Reduction of *Tgfbr1* in *Gli1-CreER;Arid1b^fl/fl* mice rescues the impairment of GLI1+ population and tissue homeostasis. a–i** The reduction of *Tgfbr1* rescues the phenotypes observed in *Gli1-CreER;Arid1b^fl/fl* mouse incisors. MicroCT (**a**, **d**, **g**), HE staining (**b**, **e**, **h**), and *Dspp* (red) (**c**, **f**, **i**) in situ hybridization of incisors from control (**a–c**), *Gli1-CreER;Arid1b^fl/fl*(**d–f**), and *Gli1-CreER;Arid1b^fl/fl;Tgfbr1^fl/+* (**g–i**) mice at 3months post-tamoxifen induction. White arrow indicates the narrowed dental pulp. Boxes in (**b**, **e**, **h**) are shown at higher magnification on the right. Yellow arrows indicate the initiation of odontoblast polarization. White two-way arrows indicate the dentin thickness. Black asterisk indicates stacked and distorted dentin. Yellow dotted lines show the distance between the cervical loop bending point and the initiation of the odontoblast. Unfilled arrows indicate the distance of the yellow dotted lines. *n* = 3. **j–o** Immunostaining of β-gal (green) in *Gli1-LacZ* (**j**, **m**), *Gli1-CreER;Arid1b^fl/fl;Gli1-LacZ* (**k**, **n**), and *Gli1-CreER;Arid1b^fl/fl;Tgfbr1^fl/+;Gli1-LacZ* (**l**, **o**) mouse incisors. **m**, **n**, **o** represent high-magnification images of the boxes in (**j–l**), respectively. White dotted lines outline the cervical loop. Yellow arrows indicate positive cells. *n* = 3. **p** Quantification of the GLI1+ cells in the dental mesenchyme of incisors from *Gli1-LacZ*, *Gli1-CreER;Arid1b^fl/fl;Gli1-LacZ*, and *Gli1-*

*CreER;Arid1b^fl/fl;Tgfbr1^fl/+;Gli1-LacZ* mice. Data are mean ± SEM, *n* = 3, unpaired two-tailed Student's t-test. *Gli1-LacZ* vs *Gli1-CreER;Arid1b^fl/fl;Gli1-LacZ*, *p* = 0.0102; *Gli1-CreER;Arid1b^fl/fl;Gli1-LacZ* vs *Gli1-CreER;Arid1b^fl/fl;Tgfbr1^fl/+;Gli1-LacZ*, *p* = 0.013. **q** Western blot of ARID1B, p-JNK, p-COFILIN, p-ERK, and β-ACTIN in the proximal incisor mesenchyme from control (C), *Gli1-CreER;Arid1b^fl/fl* (M), and *Gli1-CreER;Arid1b^fl/fl;Tgfbr1^fl/+* (T) mice. **r–w** Immunostaining of β-gal (green) in *Gli1-LacZ* (**r**, **s**), *Gli1-CreER;Arid1b^fl/fl;Gli1-LacZ* (**t**, **u**), and *Gli1-CreER;Arid1b^fl/fl;Erk2^fl/+;Gli1-LacZ* (**v**, **w**) mouse incisors. **s**, **u**, **w** represent high-magnification images of the boxes in (**r**, **t**, **v**), respectively. White dotted lines outline the cervical loop. Yellow arrows indicate positive cells. *n* = 3. **x** Quantification of GLI1+ cells in the dental mesenchyme of incisors from *Gli1-LacZ*, *Gli1-CreER;Arid1b^fl/fl;Gli1-LacZ*, and *Gli1-CreER;Arid1b^fl/fl;Erk2^fl/+;Gli1-LacZ* mice. Data are mean ± SEM, *n* = 3, unpaired two-tailed Student's t-test. *Gli1-LacZ* vs *Gli1-CreER;Arid1b^fl/fl;Gli1-LacZ*, *P* = 0.0037; *Gli1-CreER;Arid1b^fl/fl;Gli1-LacZ* vs *Gli1-CreER;Arid1b^fl/fl;Erk2^fl/+;Gli1-LacZ*, *p* = 0.0055. Source data are provided as a Source Data file. Scale bars: 2 mm (**a**, **d**, **g**); 100 μm for the rest of the images.

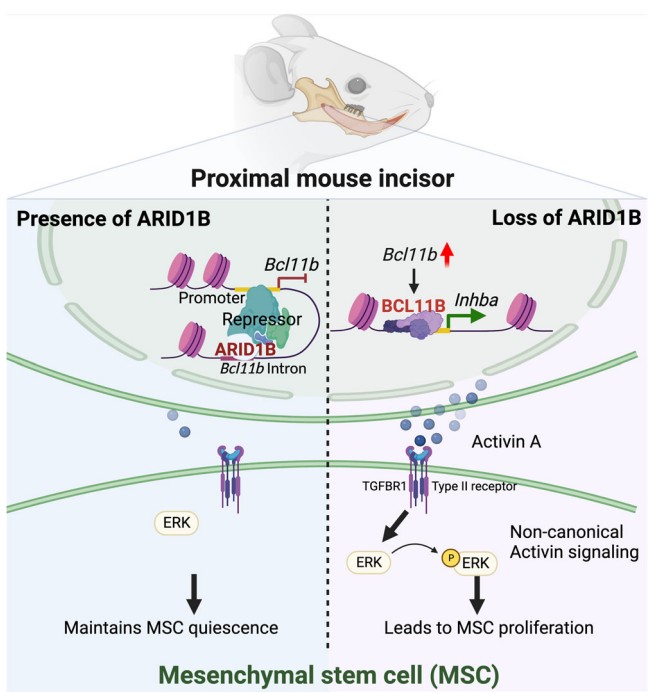

**Fig. 8 | Schematic drawing of ARID1B regulatory network within MSCs in the mouse incisor.** ARID1B directly suppresses the expression of *Bcl11b* in the proximal region of the mouse incisor to maintain MSC quiescence. After loss of *Arid1b*, *Bcl11b* is upregulated. BCL11B, as a mediator, directly regulates the activin A subunit gene (*Inhba*) expression. Activin A ectopically activates the non-canonical Activin signaling p-ERK in the MSC region and disturbs MSC homeostasis. Illustration created with BioRender.com released under a Creative Commons Attribution-NonCommercial-NoDerivs 4.0 Internatioanl license.

into this regulatory mechanism, revealing that BCL11B directly controls the expression of *Inhba*, a subunit gene of activin A, thereby activating the non-canonical Activin signaling pathway p-ERK via TGFBR1-associated receptors within MSCs in the mouse incisor.

The TGF-β signaling superfamily has been extensively investigated throughout various stages from embryonic development to adulthood, and it is known to play crucial roles in processes such as cell proliferation, differentiation, metabolism, and apoptosis. Despite its recognized significance, the precise role of the Activin signaling pathway in regulating the self-renewal, proliferation, or differentiation of adult stem cells remains elusive, particularly with respect to its non-canonical pathways. Interestingly, recent reports have indicated the potential involvement of Activin signaling in these processes, highlighting the need for further investigation. Activin has also emerged as a key regulator in the fate commitment of embryonic and cancer stem cells, as well as adult tissue stem cells[43,44]. For instance, in hair follicle stem cells, Activin signaling has been implicated in maintaining stemness, while the role of TGF-β appears to be less prominent[45,46]. In pancreatic cancer, ACTIVIN/NODAL regulate cancer stem cell self-renewal through interaction with their receptor, Alk4/7[43]. In the context of MSCs in the mouse incisor, we show a significant upregulation of the TGFBR1 receptor, and its haploinsufficiency in the *Arid1b* mutant model completely rescues the defect in tissue homeostasis and restores the MSC population. Considering the shared receptors and transduction proteins involved in both the TGF-β and Activin signaling, we propose a mechanism in which Activin A interacts with type II Activin receptors (ActRII/ActRIIB) in MSCs. This interaction leads to the recruitment, phosphorylation, and activation of the type I receptor TGFBR1, subsequently initiating downstream signaling events. Importantly, this mechanism occurs in the absence of ARID1B. Further studies are needed to validate and provide a comprehensive

understanding of this proposed regulatory mechanism. Intriguingly, the canonical pSMAD2 pathway is exclusively activated in differentiated dental pulp cells, not in MSCs. However, the widespread expression of TGF-β ligands and receptors in both MSCs and differentiated cells suggests the potential involvement of other regulators, such as extracellular matrix proteins, in releasing latent TGF-β ligands for activation, thereby activating the pSMAD2/pSMAD3 signaling pathway. Further investigation is warranted to elucidate these intricate regulatory mechanisms. Importantly, our study demonstrates how the ARID1B-BCL11B regulatory network may be specifically responsible for regulating the non-canonical Activin signaling to control the fate of MSCs in maintaining tissue homeostasis.

Our study shows definitive and compelling evidence for the crucial role of ARID1B in regulating MSC homeostasis by suppressing non-canonical Activin signaling. This regulatory mechanism involves BCL11B, which is controlled by ARID1B and acts at the promoter to regulate the expression of the activin A subunit, *Inhba*. Elevated *Inhba* expression leads to the ectopic deposition of activin A, which in turn activates p-ERK signaling in MSCs, resulting in changes in MSC fate and disruption of tissue homeostasis. These findings highlight the cellular function of ARID1B in controlling MSC fate and maintaining tissue homeostasis, underscoring the significance of the ARID1B-BCL11B-non-canonical Activin signaling regulatory network in controlling the fate of MSCs during tissue homeostasis.

## Methods

### Animals
This study was conducted in compliance with the regulations and guidelines set forth by the Institutional Animal Care and Use Committee at the University of Southern California. The mice were maintained in a pathogen-free facility and appropriate measures were taken to ensure their welfare. At the end of the study, humane euthanasia was carried out using carbon dioxide overdose followed by cervical dislocation.

This study used and cross-bred the following mouse lines: *Arid1b*$^{fl/fl}$ (JAX:032061)[13], *Bcl11b*$^{fl/fl}$ (JAX:034469)[47], *Tgfbr1*$^{fl/fl}$ (JAX: 028701)[48], *Erk2*$^{fl/fl}$ (JAX:019112)[49], *Gli1-CreER* (JAX:007913)[50], *Sox2-CreER* (JAX: 017593)[51], and *Gli1-LacZ* (JAX:008211)[52].

### Tamoxifen administration
To prepare the injection agent, tamoxifen (Sigma-Aldrich, T5648) and corn oil (Sigma-Aldrich, C8267) were mixed to achieve a concentration of 20 mg/mL. We administered intraperitoneal injections of tamoxifen to mice at one month of age, at a dosage of 1.5 mg per 10 g of body weight, given for three consecutive days (one injection per day). The control mice utilized in our study were *Arid1b*$^{fl/fl}$ mice subjected to the same tamoxifen injection protocol.

### Notch movement assay
After one week of tamoxifen induction, control and *Gli1-CreER;Arid1b*$^{fl/fl}$ mice were anesthetized and a notch was created on the enamel of one incisor above the gumline using a carbide bur (Brasseler USA, 018554U0). The amount of incisor growth was measured every other day after the notch was made.

### Histological analysis
The mouse mandibles were carefully dissected and fixed in 4% paraformaldehyde (PFA) at 4 °C overnight. Subsequently, the samples were decalcified in 10% EDTA/PBS solution for 2–4 weeks. After decalcification, the mandibles were subjected to a series of ethanol and xylene baths for dehydration and subsequently embedded in paraffin. The tissue samples were sliced into 4 μm sections using a microtome (Leica, RM2235) and then subjected to Hematoxylin and Eosin (HE) staining for histological analysis. The HE staining procedure was carried out according to standard protocols.

## Immunofluorescence and in situ hybridization

The mouse mandibles were prepared for immunofluorescence and in situ hybridization analysis following the protocol described above. Briefly, the mandibles were dehydrated in sucrose/PBS solutions of increasing concentrations (15%, 30%, and sucrose/O.C.T.), before being embedded in O.C.T. compound. The samples were sliced into 8 μm sections with a cryostat (Leica, CM1850) and collected onto slides. For immunofluorescence staining, the slides were incubated in a blocking solution (PerkinElmer, FP1012) for 1 h at room temperature before overnight incubation with primary antibodies at 4 °C. After overnight primary antibody incubation, the slides were rinsed with PBST and then incubated with Alexa-conjugated secondary antibodies (1:200, Invitrogen) for 2 h at room temperature, counterstained with DAPI (Thermo Fisher Scientific, 62248), and then mounted. All the images were captured with a fluorescence microscope (Keyence, BZ-X810).

The primary antibodies used in this study were: ARID1B (1:100, Abcam, ab244351), Ki67 (1:100, Abcam, ab15580), β-galactosidase (β-GAL) (1:100, Abcam, ab9361), Activin A (1:100, R&D Systems, MAB3381), p-SMAD2 (1:100, Cell Signaling, 18338), p-COFILIN (1:100, Cell Signaling, 3313), p-JNK (1:100, Cell Signaling, 9255), and p-ERK (1:100, Cell Signaling, 4370).

in situ hybridization was performed using the RNAscope Multiples fluorescent v2 kit (Advanced Cell Diagnosis, 323100) according to the manufacturer's instructions. The probes used in this study were synthesized by Advanced Cell Diagnostics: Probe-Mm-*Dspp* (448301), Probe-Mm-*Bcl11b*-C3 (413271-C3), Probe-Mm-*Inhba* (455871), Probe-Mm-*Tgfbr1* (406201), Probe-Mm-*Fmod* (479421), Probe-Mm-*Pax9* (454321), Probe-Mm-*Thsd7b* (850281-C3), Probe-Mm-*Adamts18* (452251).

## EdU incorporation, staining, and TUNEL assays

For EdU labeled LRCs, *Gli1-CreER;Arid1b^{fl/fl}* and control mice were injected intraperitoneally with 5-ethynyl-2′-deoxyuridine (EdU) at a dosage of 25 μg/g body weight once per day from postnatal day 5 until one month of age. One month after the final EdU injection, tamoxifen injections were administered once daily for three consecutive days. Mandibles were collected one week after the last tamoxifen injection, and then dissected, fixed, and decalcified as described above for immunofluorescence and in situ hybridization to prepare for cryosectioning. EdU staining was detected using the Click-iT plus EdU cell proliferation kit (Thermo Fisher Scientific, C10637). Cell apoptosis was detected using the Click-iT plus TUNEL Assay kit (Thermo Fisher Scientific, C10617). Both procedures were performed according to the manufacturer's instructions.

## Fluorescent double labeling

To visualize newly forming dentin, we used calcein and Alizarin red S, which produce green and red fluorescence, respectively. Calcein was dissolved in PBS at a concentration of 4 mg/ml, while Alizarin red S was dissolved in bacteriostatic water at 8 mg/ml and filtered through a Millipore filter. Calcein was injected intraperitoneally one week after tamoxifen induction, and Alizarin red S was injected five days later at a dosage of 20 mg/kg body weight. Two days after Alizarin red S injection, the mouse incisor was collected, and fluorescent complexes bound to calcium were visualized using a fluorescence microscope (Keyence, BZ-X810).

## RNA-sequencing analysis

After four days of tamoxifen induction, the proximal mesenchymal tissue of the mouse incisor was dissected from *Gli1-CreER;Arid1b^{fl/fl}* and control mice. RNA was extracted using the RNeasy Micro Kit (Qiagen, 74004) as per the manufacturer's instructions. The cDNA library was prepared, and sequencing was carried out by the Molecular Genomics Core at the University of Southern California (USC). Raw data were trimmed, aligned to the mouse mm10 genome, and subjected to differential expression analysis using Partek Flow, with a significance of FDR ≤ 0.1 and absolute fold change > 1.5.

## Mesenchymal cell isolation

The mouse incisor proximal mesenchymal tissue was harvested and mechanically fragmented into small pieces to aid in cell digestion using 2 mg/mL collagenase type I (Worthington, LS004196) and 4 mg/mL dispase II (Sigma-Aldrich, 54905400) dissolved in α-MEM media (Gibco, 12571063). The cells were incubated on a thermomixer (Eppendorf) at 37 °C with 44 x g shaking for 15 min, after which the digestion was terminated by adding α-MEM media with 10% FBS (Gibco, 16000044). The sample was then centrifuged at 600 x g for 5 min, the supernatant was discarded, and the pellet was resuspended in α-MEM media with 10% FBS.

## Single-cell RNA sequencing

A single-cell RNA sequencing was performed for control and *Gli1-CreER;Ardi1b^{fl/fl}* mouse incisors, by employing pooled samples of eight mouse incisors for each group. After four days of tamoxifen induction, the mouse incisor proximal mesenchymal tissue was collected from *Gli1-CreER;Arid1b^{fl/fl}* and control mice, and the cells were isolated as described in the "Mesenchymal cell isolation" section. To prepare for single-cell RNA sequencing, single cell suspensions from each sample were loaded onto the 10X Chromium system and were barcoded using the Chromium Next GEM Single Cell 3′ v3.1. The libraries were constructed following the protocol of the Chromium Next GEM Single Cell 3' Reagent Kits User Guide (v3.1 Chemistry) and were sequenced using the Illumina Novaseq System at the Technology Center for Genomics and Bioinformatics at the University of California, Los Angeles (UCLA). The quality control, mapping, and count table assembly of the libraries were processed using the CellRanger pipeline against the mouse mm10 annotated genome sequence. The raw read counts obtained from the CellRanger count command were analyzed using the Seurat 4.0 R package[53]. The analyzed cell numbers were 9210 for the control and 8915 for the *Gli1-CreER;Arid1b^{fl/fl}* samples.

## Single-cell ATAC sequencing

A single-cell ATAC sequencing was performed for the control and *Gli1-CreER;Ardi1b^{fl/fl}* mouse incisors, by employing pooled samples of eight mouse incisors for each group. After four days of tamoxifen induction, the mouse incisor proximal mesenchymal tissue was collected from *Gli1-CreER;Arid1b^{fl/fl}* and control mice, and the cells were isolated as described in the "Mesenchymal cell isolation" section. Then the cells were treated with lysis buffer to release nuclei. Single-cell ATAC-seq libraries were prepared using the Chromium Next GEM Single Cell ATAC Library & Gel Bead Kit (10x Genomics, PN-1000176). Briefly, the single-nuclei suspension was loaded onto the 10X Chromium system and barcoded for single-cell ATAC sequencing. The libraries were constructed according to the Chromium Single Cell ATAC Library & Gel Bead Kit protocol (v1.1 Chemistry). The scATAC-seq libraries were sequenced using the Illumina NovaSeq System at the Technology Center for Genomics and Bioinformatics at the University of California, Los Angeles (UCLA). The raw data was preprocessed, filtered, and mapped to the reference genome using the CellRanger ATAC pipeline against the mouse mm10 annotated genome sequence, and then used Signac for analyzing clusters and calling peaks[54].

## CRISPRi vector design and transfection

A gRNA sequence of CAATCTTTCACGGCTACCCG with PAM sequence of TGG was designed based on its proximity to the ARID1B binding site (+150 bp to −150 bp). This gRNA sequence was then cloned into the pCas-Guide-Puro-CRISPRi plasmid (GE201122B, Origene Technologies) by Origene Technologies, resulting in the generation of the *Bcl11b* CRISPRi vector. As a control, pCas-Guide-Puro-CRISPRi-Scramble vector (GE100084, Origene Technologies) was used.

Primary cells were isolated as described in the "Mesenchymal cell isolation" section and transfected with the CRISPRi vectors using Lipofectamine LTX reagent (15338030, Invitrogen). The transfection procedure followed the protocol provided with the Lipofectamine® LTX DNA transfection reagents. Following transfection, RNA was collected, and reverse transcribed into cDNA for quantitative PCR (qPCR) analysis of *Bcl11b* expression. The following primer sequences were obtained from PrimerBank (Wang et al., 2012) and used for qPCR reactions: *Bcl11b* (Forward: 5′-CCTCCGTGATTACTTCACCTCT-3′; Reverse: 5′-TGACCCTCACCCTGAGTCC-3′).

## Western blotting
The incisor proximal mesenchyme from control and *Gli1-CreER;Arid1b^{fl/fl}* mice was collected and lysed in RIPA buffer (Cell Signaling, 9806) with protease inhibitor (Thermo Fisher Scientific, 1861278) for 30 min at 4 °C. The soluble fraction was obtained by centrifugation at 14,000 × g at 4 °C for 10 min. Total protein extracts were then loaded onto a 4–15% precast polyacrylamide gel (Bio-Rad, 456-1084) and transferred onto PVDF membranes (Millipore, ISEQ00005). The membranes were subsequently blocked with 5% nonfat milk for 1 h before being incubated with primary antibodies overnight at 4 °C. After that, the horseradish-peroxidase (HRP)-conjugated secondary antibodies were used to detect the proteins bound with primary antibodies. The western blot images were visualized using Azure 300 imaging system (Azure Biosystems) and the protein band intensities were analyzed in ImageJ.

The primary antibodies used for western blotting in this study were: anti-ARID1B (Abcam, ab244351, 1:1000), anti-p-SMAD2 (Cell Signaling, 18338, 1:1000), anti-SMAD2 (Cell Signaling, 5339, 1:1000), anti-p-AKT(Cell Signaling, 4060, 1:1000), anti-AKT (Cell Signaling, 9272, 1:1000), anti-p-P38 (Cell Signaling, 4511, 1:1000), anti-P38 (Cell Signaling, 8690, 1:1000), anti-p-ERK (Cell Signaling, 4370, 1:1000), anti-ERK (Cell Signaling, 4695, 1:1000), anti-p-JNK (Cell Signaling, 9255, 1:1000), anti-JNK (Cell Signaling, 9252, 1:1000), anti-p-NF-κB (Cell Signaling, 3033, 1:1000), anti-NF-κB (Cell Signaling, 8242, 1:1000), anti-p-COFILIN (Cell Signaling, 3313, 1:1000), anti-COFILIN (Cell Signaling, 5175, 1:1000), and anti-β-ACTIN (Abcam, ab20272, 1:1000).

## CHIP-qPCR
The mouse incisor proximal mesenchymal tissue was collected from control mice, and the cells were isolated as described in the "Mesenchymal cell isolation" section. The ARID1B and BCL11B CHIP procedure was performed according to the protocol for the CHIP-IT Express Kit (Active motif, 53008). The antibodies used for ARID1B CHIP were 8 µl of abcam ab57461 and 5 µl of Cell Signaling 92964. The antibody used for BCL11B CHIP was 10 µl of Cell Signaling 12120. The antibody used for IgG CHIP was 5 µl of Cell Signaling 3900. Standard protocol qPCR reaction was run using SsoFast EvaGreen Supermix (Bio-Rad, 172-5202) on a Real-Time Systems (Bio-Rad, CFX96). The sequences of qPCR primers used to detect enriched DNA fragments for ARID1B binding site were 5′-ACAGTGTGTTGCAAACCAGG-3′ (forward) and 5′-TACACAAAGGGCATCCCCAG-3′ (reverse). The sequences of qPCR primers used to detect enriched DNA fragments for BCL11B binding site were 5′-CCCTGAGTTATCAGCAGCTTGTC-3′ (forward) and 5′-TCCCTGAAGCAAGGAGACAGG-3′ (reverse).

## Mouse incisor explant culture
The proximal ends of the incisors were collected and dissected from *Gli1-lacZ*and *Gli1-CreER;Arid1b^{fl/fl};Gli1-LacZ* mice at 5 days post-induction, and cultured with a Trowell culture system in vitro. BGjb medium with 10% fetal bovine serum (Gibco), 2% penicillin/streptomycin (Invitrogen), and 0.1 mg/ml ascorbic acid (Sigma) were used to culture the explant tissue. Affi-Gel blue agarose beads (BioRad) were prewashed in PBS and soaked in IgG control (0.2 µg/µl, R&D system) and ACTIVIN A neutralizing antibody (0.2 µg/µl, Abcam, ab89307) at 37 °C

for at least 1 h. Then the beads were placed around the proximal end of the incisor. The explanted incisors were harvested after 3 days of cultivation and fixed in 4% paraformaldehyde at 4 °C overnight. Afterward, explant samples were processed with regular dehydration, sectioning, and immunostaining protocols.

## MicroCT analysis
The mouse incisors were collected and fixed with 4% paraformaldehyde for 24 h before being transferred to PBS for microCT imaging. Radiological images were captured using a SCANCO µCT50 (Scanco V1.28) at the University of Southern California Molecular Imaging Center with a resolution of 10 µm, and X-ray source at 90 kVp and 78 mA.

## Statistical analysis
Statistical analysis was conducted using GraphPad Prism9, with paired Student's t-tests used to determine significance. Data are presented as mean ± SEM, and $p < 0.05$ was considered statistically significant. All the measurements were taken from distinct samples.

## Reporting summary
Further information on research design is available in the Nature Portfolio Reporting Summary linked to this article.

## Data availability
The scRNA-seq, scATAC-seq, and bulk RNA-seq data generated in this study have been deposited in the GEO database under accession code GSE237305. Source data are provided with this paper.

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

## Acknowledgements

We acknowledge Dr. Bridget Samuels for critical reading and editing of the manuscript, Juechen Yang from Cincinnati Children's Hospital Medical Center and Kuo-Chang (Ted) Tseng from USC Stem Cell for technical support with scRNA-seq and scATAC-seq analysis, USC Libraries Bioinformatics Service for assisting with data analysis training, and the USC Office of Research and the Norris Medical Library for providing bioinformatics software and computing resources. This study was supported by grant funding from the National Institute of Dental and Craniofacial Research, National Institutes of Health (R01 DE025221 to Y.C.).

## Author contributions

M.Z. and Y.C. designed the study; M.Z. carried out most of the experiments and data analysis; T.G., F.P., J.F., J.J., T.Y., J.D. and P.S. participated in sample collection. J.X. provided experimental investigation. T.-V.H. participated in the microCT analysis. M.Z. and Y.C. co-wrote the paper. Y.C. supervised the research.

## Competing interests

The authors declare no competing interests.
