## [Peer Review File · Nature Communications]

ARID1B maintains mesenchymal stem cell quiescence via inhibition of BCL11B-mediated non-canonical Activin signalingReviewer #1 (Remarks to the Author):

In the manuscript "ARID1B maintains mesenchymal stem cell quiescence via inhibition of BCL11B-mediated non-canonical Activin signaling" by Zhang et al. investigated the function and underlying molecular mechanism of ARID1B as a critical epigenetic modifier for mouse incisor MSC quiescence and tissue homeostasis.

Using mouse incisor model for a continuously renewing organ, the authors incorporated scRNA-seq and scATAC-seq bioinformatics analysis and histological assessment of incisor after Gli1(+) MSC lineage-specific KO of Arid1b.

Based on their findings from a series of cKO and subsequent rescue experiments, the authors propose that ARID1B keeps Gli (+) MSC from unchecked/premature proliferation by specifically inhibiting the BCL11B-Inhba-ERK axis, thereby contributes to mouse incisor growth and tissue homeostasis.

The most interesting aspect of the study includes their finding that expression of BCL11B, a BAF subunit, is directly suppressed by another BAF subunit ARID1B as a part of the molecular mechanism to epigenetically regulate incisor MSC homeostasis.

The work makes an original contribution to our knowledge about the role of BAF complex in stem cell biology, as well as extends the authors' previous findings (Du et al (2021) Development) on the role of ARID1A in regulating the fate of TAC for incisor tissue homeostasis.

Overall, this manuscript is clearly written, presenting well-structured experiments and solid/compelling data. Biological interpretation and conclusions drawn from the bioinformatics analyses were followed up by a series of functional experiments. Methodology to support their conclusions is relevant and well-reasoned. The methods section provided enough detail for the work to be reproduced.

**Below I provide my comments and suggestions for improving the manuscript:

-Although briefly commented in the Discussion section of the manuscript, data is not presented on what happens to ARID1A when ARID1B is deleted in the Gli(+) MSC. In a previously published work, the authors presented findings that ARID1A promotes the mitotic exit of TACs in mouse incisor. If so, any changes in ARID1A expression observed in the Arid1b cKO mice? Any compensation for partial loss of ARID1B with ARID1A in Arid1b cKO?

-It would be more convincing if data on ARID1B deletion efficiency in Gli1-CreER(+);Arid1b(f/f) mice is provided (e.g. ARID1B immunohistology).

-It may be of interest to understand the mechanism of BCL11B activation. How does ERK activation affect PTM (post-translational modification) of BCL11B, of which activity is known to be highly regulated by phosphorylation?

-The major drawback in the data presented in Figure 7 lies in the indirect evidence on non-canonical ACTIVIN signaling for ERK activation to promote MSC proliferation. The rescue experiment with Tgfbr1 cKO may affect both canonical and non-canonical pathways downstream of the receptor upon ACTIVIN engagement. It may be insufficient to support their argument that ACTIVIN signaling for non-SMAD pathway leading to ERK activation is the responsible pathway for ARID1B-mediated suppression of MSC proliferation. Would a ligand trap that binds Activin A (e.g. STM 434) give rise to the similar effect as in Tgfbr1 cKO?

-It is unclear what the control mice are. Gli1-CreER(-);Arid1b(f/f) with TMX injection? or Gli1-CreER (+); Arid1b (+/+)?

-Line 130, I think the authors meant to say "non-canonical Activin signaling....", instead of "non-Activin signaling....."

-It is known that ARID1B can bind to intron region of target genes. In that regard, this study provides another example of intronic regulation by ARID1B-containing BAF complex. It would be a clearer demonstration, if the authors include CRISPRi data of the promoter region of Bcl11b.

-Line 725, ARID1A CHIP should be ARID1B CHIP.

Reviewer #2 (Remarks to the Author):

In this manuscript, Zhang etc. have reported a potential role of Arid1b in incisor mesenchymal cells by focusing on the mesenchymal stem cells population. This is part of series of studies from Professor Chai's lab, using Gli1 as the molecular tool to tackle incisor mesenchymal stem cells. Comparing with the previous publications using Gli1 Cre ER, the data presented have significant improvement regarding quality control, imaging manipulation and logic presentation. However, critiques still persist for the various models and data analysis etc. inside the manuscript, which potentially caused false interpretation of the results hence the logic, therefore making it difficult to judge the credibility of the claims made in the paper.

The current major point:

For Gli1-CreER x Arid1b flox/flox mice, the correct control should be Gli1-CreER x Arid1b flox/flox mutant itself but without tamoxifen induction. It looks the authors used other strain as the control instead, which is a key issue. As we all know cells carrying Cre construct do receive some defects due to leaking of Cre recombinase production although the ER system persists. Therefore it is not possible to judge if the phenotypes observed (Figure 1, Figure 2 and figure 3) are due to Arid1b deletion or because of the Cre recombinase toxicity.

Only after the authors clarify which kind of control(s) they have used in the paper I am willing to continue my review.

Reviewer #3 (Remarks to the Author):

This manuscript examines the role of ARID1B in stem cell development. Using the rodent mouse incisor as a model, the authors illustrate that ARID1B controls mesenchyme stem cell (MSC) proliferation by inhibiting Bcl11b expression; this potentially occurs through ARID1B binding to an intronic element in the Bcl11b gene. Reducing Activin A signaling or Bcl11b restores the MSC compartment by preventing aberrant proliferation. This study relies on single cell RNA sequencing and scATAC sequencing, and at several places uses genetically modified mutants to support their hypotheses. Overall, the quality of the data is exceptional and supports the authors' conclusions. These findings represent a major step forward in understanding how regulation of the stem cell niche in the rodent incisors is controlled and may reflect on a new mechanism to control aberrant MSC proliferation in other stem cell niches as well. While the study is strong, there are a few points that need to be addressed.

1. In the introduction and discussion, the role of ARID1B in stem cell homeostasis is described in very general terms. This implies that ARID1B-based regulation is a conserved mechanism that functions in all stem cell compartments. In fact, the study eloquently shows that ARID1B plays a role in maintaining stem cell homeostasis in the cervical loop of the developing mouse incisor. While it is reasonable that ARID1B plays a similar role in other MSC compartments, the authors do not show proof of this. Therefore, their statements of a global role for ARID1B, made by not specifically stating that they are only looking at the mouse incisor, is misleading. The authors need to clearly state what their data show versus what it implies.

2. In Figure 1L to 1W, the authors examine changes in the cervical loop of the mouse incisor in *Arid1b* conditional mutants. However, none of the measurements are quantified. While the changes look relatively clear, the author should consider adding some sort of quantitation to the study. Alternatively, the authors should simply be clear and say these are qualitative changes. Further, the decreased distance of *Dspp* expression in mutants could reflect a smaller cervical loop, something the authors should at least address.

3. In Figure 2N-R, the authors show co-labeling of Ki67 and Gli-LacZ. Statistically, the changes are significant, with about one co-labeled cell in the control cervical loop and about eight in the mutant cervical loop. However, this is out of apparently hundreds and hundreds of cells. This seems a little bit like a misuse of statistics, in which there is a statistically significant change among co-labeled cells, but not necessarily a significant change in the entire population. The authors need to clearly state how such a small change can lead to later changes shown in Figure 1.

4. In Figure 2, the authors show a single cell RNA sequencing UMAP. The authors then re-cluster the dental mesenchyme cells to illustrate the sub populations of the dental mesenchyme cells. However, in Figure 3, the expression of *Bcl11b* and *Gli1* are both shown on the original UMAP. This makes it very difficult to determine where within the reclustered populations this expression occurs. This is important, since the authors then use violin plots to illustrate the expression levels in the different dental mesenchyme clusters. They need to show the expression of *Bcl11b* and *Gli1* in the re-clustering to make the violin plots believable.

5. In Figure 2E-J, the authors show that loss of *Arid1b* leads to an expansion of *Bcl11b*. However, quantification of expression is not shown, even though they're using RNAscope. To a reviewer, it looks as if the expression is higher, but not more widespread as the authors indicate with their increased number of yellow arrows. The authors need to clarify whether they believe there's an expansion of *Bcl11b* expression or simply an up regulation of expression in cells already expressing *Bcl11b* and how they can differentiate between the two.

6. The authors use single nucleotide ATAC-seq to examine differential peaks between control and mutant MSC's. Identifying a region in the intron of *Bcl11b*, the authors show that this region is enriched in ARID1B and that *Bcl11b* expression is reduced when the intronic region is targeted by a CRISPR guide. While these findings suggest that ARID1B downregulates *Bcl11b* expression, these are in vitro assays. If the authors want to prove that this is region is involved in a functional control of MSC development, they must produce in vivo evidence that this enhancer drives expression of a transgene in MSC populations and is down regulated by ARID1B.

Minor comments:

1. The introduction could be shortened. There is too much review of every aspect covered in this manuscript. The authors need to decide what they want to cover and hit the high points.
2. In Figure 2F to J, the authors administer tamoxifen over a one month period. However, they do not state in the Materials and Methods or the Results section how this was accomplished. Was it once a day? Every other day? This needs to be explicitly stated at least in the Methods.
3. The others need to state if the single cell experiments were performed one or two times. Current accepted procedures suggest that one time is sufficient when results are used as a tool for a study like this. However, this does need to be stated.
4. In Figure 4B, there needs to be some sort of heat map legend in the figure.

January 17, 2024

NCOMMS-23-37342

Point-by-point response to the reviewers' comments:

Reviewer #1 (Remarks to the Author):

In the manuscript "ARID1B maintains mesenchymal stem cell quiescence via inhibition of BCL11B-mediated non-canonical Activin signaling" by Zhang et al. investigated the function and underlying molecular mechanism of ARID1B as a critical epigenetic modifier for mouse incisor MSC quiescence and tissue homeostasis.

Using mouse incisor model for a continuously renewing organ, the authors incorporated scRNA-seq and scATAC-seq bioinformatics analysis and histological assessment of incisor after Gli1(+) MSC lineage-specific KO of Arid1b.

Based on their findings from a series of cKO and subsequent rescue experiments, the authors propose that ARID1B keeps Gli (+) MSC from unchecked/premature proliferation by specifically inhibiting the BCL11B-Inhba-ERK axis, thereby contributes to mouse incisor growth and tissue homeostasis.

The most interesting aspect of the study includes their finding that expression of BCL11B, a BAF subunit, is directly suppressed by another BAF subunit ARID1B as a part of the molecular mechanism to epigenetically regulate incisor MSC homeostasis.

The work makes an original contribution to our knowledge about the role of BAF complex in stem cell biology, as well as extends the authors' previous findings (Du et al (2021) Development) on the role of ARID1A in regulating the fate of TAC for incisor tissue homeostasis.

Overall, this manuscript is clearly written, presenting well-structured experiments and solid/compelling data. Biological interpretation and conclusions drawn from the bioinformatics analyses were followed up by a series of functional experiments. Methodology to support their conclusions is relevant and well-reasoned. The methods section provided enough detail for the work to be reproduced.

We appreciate the reviewer's positive feedback on our study and valuable suggestions for improving our manuscript. Below, you will find our responses to address all the questions point-by-point.

****Below I provide my comments and suggestions for improving the manuscript:**

-Although briefly commented in the Discussion section of the manuscript, data is not presented on what happens to ARID1A when ARID1B is deleted in the Gli(+) MSC. In a previously published work, the authors presented findings that ARID1A promotes the mitotic exit of TACs in mouse incisor. If so, any changes in ARID1A expression observed in the *Arid1b* cKO mice? Any compensation for partial loss of ARID1B with ARID1A in *Arid1b* cKO?

We thank the reviewer for bringing up this point. We observed an expanded expression of ARID1A in *Arid1b* cKO mouse incisors compared to the control, particularly in the proximal dental follicle and dental pulp cells (Response Figure 1A-1D).

To explore the role of ARID1A in *Arid1b* cKO mice, we collected incisor samples of *Gli1-CreER;Arid1b^{fl/fl};Arid1a^{fl/+}* mice 3 months post-induction. We observed more severe impairment of tissue homeostasis in the incisors of *Gli1-CreER;Arid1b^{fl/fl};Arid1a^{fl/+}* mice through HE staining, including stacked dentin and disorganized odontoblasts (Response Figure 1E-1J). These findings strongly imply that ARID1A may partially compensate for the function of ARID1B.

Response Figure 1. ARID1A is essential in regulating mouse incisor tissue homeostasis in the *Gli1-CreER;Arid1b^{fl/fl}* mouse. (A-D) Immunostaining of ARID1A in the incisors from control (A, B) and *Gli1-CreER;Arid1b^{fl/fl}* (C, D) mice 1 week post-tamoxifen induction. B and D represent the high-magnification images of the boxes in A and C, respectively. The white dotted lines outline the cervical loop. Yellow arrows point to the positive cells. (E-J) HE staining of incisors from control (E, H), *Gli1-CreER;Arid1b^{fl/fl}* (F, I), and *Gli1-CreER;Arid1b^{fl/fl};Arid1a^{fl/+}* (G, J) mice at 3 months after tamoxifen induction. H, I, and J represent the high-magnification images of the dashed line boxes in E, F, and G. Yellow arrows indicate the initiation of odontoblast polarization. Black asterisks indicate stacked and distorted dentin. Scale bars: 100 μ m.

-It would be more convincing if data on ARID1B deletion efficiency in *Gli1-CreER(+);Arid1b(f/f)* mice is provided (e.g. ARID1B immunohistology).

We thank the reviewer for the comment. We included the immunostaining of ARID1B in control and *Gli1-CreER;Arid1b^{fl/fl}* mouse incisors in Supplementary Figure 1A-1D. ARID1B was efficiently deleted at 5 days after tamoxifen induction.

Supplementary Fig. 1

-It may be of interest to understand the mechanism of BCL11B activation. How does ERK activation affect PTM (post-translational modification) of BCL11B, of which activity is known to be highly regulated by phosphorylation?

We thank the reviewer for these comments. In our study, we discovered that BCL11B directly regulates the ligand of Activin signaling, consequently activating the non-canonical p-ERK signaling pathway. We acknowledge the intriguing prospect of exploring the post-translational modification (PTM) of BCL11B through ERK activation. Delving into this aspect is beyond the scope of our current study, but it's definitely a promising direction for future research.

-The major drawback in the data presented in Figure 7 lies in the indirect evidence on non-canonical ACTIVIN signaling for ERK activation to promote MSC proliferation. The rescue experiment with *Tgfb1* cKO may affect both canonical and non-canonical pathways downstream of the receptor upon ACTIVIN engagement. It may be insufficient to support their argument that ACTIVIN signaling for non-SMAD pathway leading to ERK activation is the responsible pathway for ARID1B-mediated suppression of MSC proliferation. Would a ligand trap that binds Activin A (e.g. STM 434) give rise to the similar effect as in *Tgfb1* cKO?

We agree with the reviewer that it is important to validate the role of ligand Activin A in maintaining MSC homeostasis through a non-canonical ERK pathway.

Unfortunately, STM 434 is currently unavailable for commercial purchase. As an alternative validation approach, we conducted an incisor explant culture with IgG as the control group and neutralizing Activin A antibody as the treatment for control and *Arid1b* mutant mouse incisors. Notably, the organ culture results displayed that neutralizing Activin A in *Arid1b* mutant mouse incisor explants restored the population of GLI1+ MSCs

compared to the IgG treatment (Revised Figure 5N-5R). These findings strongly suggest that reducing the Activin A level itself could rescue the diminished population of Gli1+ MSCs in the *Arid1b* mutant mouse incisor.

Additionally, to confirm that the p-ERK signaling pathway is functionally downstream of Activin signaling, we compared western blots from control, *Arid1b* mutant, and *Tgfbr1* rescue groups, and found that only p-ERK signaling was restored in the rescue model (Revised Figure 7Q). Moreover, to further validate the function of the p-ERK pathway, we generated another rescue model, *Gli1-CreER;Arid1b^{fl/fl};Erk2^{fl/+}*. We compared the GLI1+MSCs among control, *Arid1b* mutant, and *Erk2* rescue groups, and found that the reduced GLI1+ cells in the *Arid1b* mutant were restored in the *Erk2* rescue model (Revised Figure 7R-7X). This result suggests that p-ERK signaling is functionally downstream following the loss of *Arid1b*. We have updated the relevant figures and text to reflect these results.

-It is unclear what the control mice are. Gli1-CreER(-);Arid1b(f/f) with TMX injection? or Gli1-CreER (+); Arid1b (+/+)?

We thank the reviewer for raising this point. We used *Arid1b^{fl/fl}* mouse with tamoxifen injections as the control. We added this information to the Materials and Methods under the tamoxifen administration section.

-Line 130, I think the authors meant to say "no-canonical Activin signaling....", instead of "non-Activin signaling....."

We thank the reviewer for the comment. We changed the wording as shown here:

“Significantly, we verified the functional significance of BCL11B and non-canonical Activin signaling downstream of ARID1B in preserving MSC homeostasis.”

-It is known that ARID1B can bind to intron region of target genes. In that regard, this study provides another example of intronic regulation by ARID1B-containing BAF complex. It would be a clearer demonstration, if the authors include CRISPRi data of the promoter region of Bcl11b.

We thank the reviewer for these comments. In response to the reviewer's suggestion, we designed two gRNAs, referred to as gRNA-PA and gRNA-PB, targeting the region upstream of the transcription start site (TSS), as depicted in Response Figure 2A. Following CRISPRi treatment of primary cells from the proximal region of the mouse incisor, we found a reduction in the *Bcl11b* expression level after *Bcl11b* promoter CRISPRi treatment, aligning with our expectations (Response Figure 2B).

In addition, we also designed an alternative gRNA, referred as gRNA-2, to target a different locus within the target intron (Response Figure 2C). We applied this gRNA to primary cells collected from the proximal mouse incisor. Subsequently, we conducted a comparison of *Bcl11b* expression levels and observed no significant expression difference between the scrambled control and the group treated with *Bcl11b* gRNA-2 (Response Figure 2D). This result strongly supports the notion that the binding region we identified serves as the functional ARID1B binding site.

Response Figure 2. CRISPRi treatment of primary mouse incisor mesenchyme cells targeting promoter and other locus. (A) Schematic illustration of gRNAs targeting the promoter of *Bcl11b*. (B) RT-qPCR analysis of *Bcl11b* expression following CRISPRi treatment targeting promoter region. (C) gRNA-2 position within the target intron. (D) RT-qPCR analysis of *Bcl11b* expression following CRISPRi treatment targeting another intron region. (gRNA-PA: AGTTACGCCGGGTTTTGCAC; gRNA-PB: ACGTGAAGATGGCGGAGTCC; gRNA-2: AACTGAACTGTAACCTGTGT).

-Line 725, ARID1A CHIP should be ARID1B CHIP.

We thank the reviewer for the comment. We changed the wording as shown here:

“The antibodies used for ARID1B CHIP were 8 μ l of Abcam ab57461 and 5 μ l of Cell Signaling 92964.”

Reviewer #2 (Remarks to the Author):

In this manuscript, Zhang etc. have reported a potential role of *Arid1b* in incisor mesenchymal cells by focusing on the mesenchymal stem cells population. This is part of series of studies from Professor Chai's lab, using *Gli1* as the molecular tool to tackle incisor mesenchymal stem cells. Comparing with the previous publications using *Gli1 Cre ER*, the data presented have significant improvement regarding quality control, imaging manipulation and logic presentation. However, critiques still persist for the various models and data analysis etc. inside the manuscript, which potentially caused false interpretation of the results hence the logic, therefore making it difficult to judge the credibility of the claims made in the paper.

The current major point:

For *Gli1-CreER x Arid1b flox/flox* mice, the correct control should be *Gli1-CreER x Arid1b flox/flox* mutant itself but without tamoxifen induction. It looks the authors used other strain as the control instead, which is a key issue. As we all know cells carrying Cre construct do receive some defects due to leaking of Cre recombinase production although the ER system persists. Therefore it is not possible to judge if the phenotypes observed (Figure 1, Figure 2 and figure 3) are due to *Arid1b* deletion or because of the Cre recombinase toxicity.

Only after the authors clarify which kind of control(s) they have used in the paper I am willing to continue my review.

We thank reviewer for these comments. To clarify, we utilized *Arid1b^{f1/f1}* mice subjected to tamoxifen injection as the control group. This information has been included in the Materials and Methods section under the tamoxifen administration subsection.

In our study, we compared *Gli1-CreER;Arid1b^{f1/f1}* mouse incisors with *Arid1b^{f1/f1}* mice post-tamoxifen injection. This approach allowed us to discern defects resulting from the loss of *Arid1b* within the GLI1+ lineage, thus reducing the influence of other potential factors on the observed phenotype. We opted not to use *Gli1-CreER;Arid1b^{f1/f1}* mice without tamoxifen induction as the control to avoid concerns about whether the phenotype was due to tamoxifen induction.

To address the reviewer's concern about Cre recombinase toxicity, we gave the same tamoxifen injection protocol to both wild type and *Gli1-CreER* mice and collected the incisors 2 months post-injection for HE staining analysis. This result showed no apparent differences between the wild type and *Gli1-CreER* mouse incisors (Response Figure 3A-3D). We also collected *Gli1-CreER;Arid1b^{fl/fl}* mouse incisor without tamoxifen injection and observed no apparent morphology differences compared to the wild type (Response Figure 3E-3F). Additionally, our prior publication demonstrated that induction of *Gli1-CreER* did not affect the GLI1+ MSC population, as evidenced in the figure below from Jing et al., 2021. These findings suggest that the Cre recombinase did not impact the mouse incisors in the current study, and the phenotype observed in the *Arid1b* mutant mice can indeed be attributed to the loss of *Arid1b* within the GLI1+ lineage. We hope this supplementary analysis effectively addresses and resolves the reviewer's concerns.

Response Figure 3. Comparison of HE staining of wild type (A, B) and *Gli1-CreER* (C-D) mouse incisors at 2 months post-tamoxifen injection, and 3-months age *Gli1-CreER;Arid1b^{fl/fl}* (E-F) mouse incisors without tamoxifen injection. B, D, and F represent high-magnification images of the boxes in A, C, and E, respectively. n=3. Scale bars: 100 μ m.

(Reference: Junjun Jing, Jifan Feng, Jingyuan Li, Hu Zhao, Thach-Vu Ho, Jinzhi He, Yuan Yuan, Tingwei Guo, Jiahui Du, Mark Urata, Paul Sharpe, Yang Chai (2021) Reciprocal interaction between mesenchymal stem cells and transit amplifying cells regulates tissue homeostasis. *eLife* 10:e59459)

Reviewer #3 (Remarks to the Author):

This manuscript examines the role of ARID1B in stem cell development. Using the rodent mouse incisor as a model, the authors illustrate that ARID1B controls mesenchyme stem cell (MSC) proliferation by inhibiting *Bcl11b* expression; this potentially occurs through ARID1B binding to an intronic element in the *Bcl11b* gene. Reducing Activin A signaling or *Bcl11b* restores the MSC compartment by preventing aberrant proliferation. This study relies on single cell RNA sequencing and scATAC sequencing, and at several places uses genetically modified mutants to support their hypotheses. Overall, the quality of the data is exceptional and supports the authors' conclusions. These findings represent a major step forward in understanding how regulation of the stem cell niche in the rodent incisors is controlled and may reflect on a new mechanism to control aberrant MSC proliferation in other stem cell niches as well. While the study is strong, there are a few points that need to be addressed.

1. In the introduction and discussion, the role of ARID1B in stem cell homeostasis is described in very general terms. This implies that ARID1B-based regulation is a conserved mechanism that functions in all stem cell compartments. In fact, the study eloquently shows that ARID1B plays a role in maintaining stem cell homeostasis in the cervical loop of the developing mouse incisor. While it is reasonable that ARID1B plays a similar role in other MSC compartments, the authors do not show proof of this. Therefore, their statements of a global role for ARID1B, made by not

specifically stating that they are only looking at the mouse incisor, is misleading. The authors need to clearly state what their data show versus what it implies.

We thank the reviewer for the comments. We modified the introduction and discussion statements according to the reviewer's suggestion.

2. In Figure 1L to 1W, the authors examine changes in the cervical loop of the mouse incisor in *Arid1b* conditional mutants. However, none of the measurements are quantified. While the changes look relatively clear, the author should consider adding some sort of quantitation to the study. Alternatively, the authors should simply be clear and say these are qualitative changes. Further, the decreased distance of *Dspp* expression in mutants could reflect a smaller cervical loop, something the authors should at least address.

We thank the reviewer for these comments. To address these concerns, we included quantification of the dental pulp cavity size and the distance of *Dspp*⁺ cells from the cervical loop in both control and *Arid1b* mutant mouse incisors at 3 months post-tamoxifen injection (Revised Figure 1R, 1S). Moreover, we revised the corresponding text to clarify the comparison of the cervical loop between control and *Arid1b* mutant mouse incisors.

3. In Figure 2N-R, the authors show co-labeling of Ki67 and Gli-LacZ. Statistically, the changes are significant, with about one co-labeled cell in the control cervical loop and about eight in the mutant cervical loop. However, this is out of apparently hundreds and hundreds of cells. This seems a little bit like a misuse of statistics, in which there is a statistically significant change among co-labeled cells, but not necessarily a significant change in the entire population. The authors need to clearly state how such a small change can lead to later changes shown in Figure 1.

We thank the reviewer for this feedback and acknowledge the concerns raised. Regarding GLI1 as an MSC marker, it's important to note that not all GLI1⁺ cells are MSCs. To address this, we compared the percentage of label-retaining cells (LRCs) engaged in proliferation between the control and *Arid1b* mutant mouse incisors. Our co-staining of LRCs with Ki67 and subsequent quantification (Revised Figure 2N-2R) revealed that approximately 6% of LRCs prematurely proliferated following the loss of *Arid1b*. This

premature proliferation might contribute to the reduction of MSCs, although we acknowledge there could be other factors contributing to this reduction.

4. In Figure 2, the authors show a single cell RNA sequencing UMAP. The authors then re-cluster the dental mesenchyme cells to illustrate the sub populations of the dental mesenchyme cells. However, in Figure 3, the expression of *Bcl11b* and *Gli1* are both shown on the original UMAP. This makes it very difficult to determine where within the reclustered populations this expression occurs. This is important, since the authors then use violin plots to illustrate the expression levels in the different dental mesenchyme clusters. They need to show the expression of *Bcl11b* and *Gli1* in the re-clustering to make the violin plots believable.

We thank the reviewer for raising this suggestion. We updated the expression of *Bcl11b* and *Gli1* in the scRNA-seq re-clustered UMAP plot in Figure 3C.

5. In Figure 2E-J, the authors show that loss of *Arid1b* leads to an expansion of *Bcl11b*. However, quantification of expression is not shown, even though they're using RNAscope. To a reviewer, it looks as if the expression is higher, but not more widespread as the authors indicate with their increased number of yellow arrows. The authors need to clarify whether they believe there's an expansion of *Bcl11b* expression or simply an up regulation of expression in cells already expressing *Bcl11b* and how they can differentiate between the two.

We thank the reviewer for this comment and have performed additional analysis to address this concern. We first took advantage of the scRNA-seq data and observed that *Bcl11b*+ cells were primarily enriched in the MSC and PDF clusters in the control group, while in the absence of *Arid1b*, there was an expansion of *Bcl11b*+ cells, for example in the dental pulp (Response Figure 4A). Furthermore, by quantifying the percentage of *Bcl11b*+ cells in the re-clustered cell population, we noted an increased number of cells expressing *Bcl11b* following the loss of *Arid1b* (Response Figure 4B). Additionally, because *Bcl11b* is expressed in *GLI1*+ cells, we isolated the *GLI1*+ cell population and conducted a comparative analysis of the average *Bcl11b* expression levels in the control and *Arid1b* mutant scRNA-seq data. This analysis revealed an enhanced *Bcl11b* expression level in cells already exhibiting *Bcl11b* expression (Response Figure 4C-4D). Moreover, to reinforce these

findings, we quantified the *Bcl11b* expression levels per cell from the *in situ* hybridization staining and updated the quantification results in Revised Figure 3K. Collectively, these results suggest that the upregulation of *Bcl11b* can be attributed to an increase in the number of cells expressing *Bcl11b* and to elevation in the expression levels within the cells that already exhibit *Bcl11b* expression.

Response Figure 4. Comparison of *Bcl11b* expression using scRNA-seq data from control and *Gli1-CreER;Arid1b^{fl/fl}* mouse incisors. (A) UMAP plot illustrating the differential expression of *Bcl11b* between control and *Gli1-CreER;Arid1b^{fl/fl}* mouse incisors. (B) Percentage of *Bcl11b*+ cells in control and *Gli1-CreER;Arid1b^{fl/fl}* mouse incisors. (C) UMAP plot displaying the expression patterns of *Gli1* and *Bcl11b* in control and *Gli1-CreER;Arid1b^{fl/fl}* mouse incisors. (D) Comparison of average *Bcl11b* expression levels in *GLI1*+ cells from control and *Gli1-CreER;Arid1b^{fl/fl}* mouse incisors.

6. The authors use single nucleotide ATAC-seq to examine differential peaks between control and mutant MSC's. Identifying a region in the intron of *Bcl11b*, the authors show that this region is enriched in ARID1B and that *Bcl11b* expression is reduced when the intronic region is targeted by a CRISPR guide. While these findings suggest that ARID1B downregulates *Bcl11b* expression, these are *in vitro* assays. If the authors want to prove that this region is involved in a functional control of MSC development, they must produce *in vivo* evidence that this enhancer drives expression of a transgene in MSC populations and is down regulated by ARID1B.

We thank the reviewer for the comment. To validate this binding activity *in vivo*, we cloned the intronic sequence into a reporter vector and incorporated it into lentivirus for injection into the proximal region of both control and *Arid1b* mutant mouse incisors (Response Figure 5A). The mice were given tamoxifen injections at 1 month of age and lentivirus (1.5 μ l per incisor, 1.03E+9 TU/mL) was slowly injected at the proximal region of mouse incisors mesenchyme at 5 days post-tamoxifen injection. The mouse incisors were collected 5 days after the lentivirus injection for analysis (Response Figure 5B). We observed the colocalization of eGFP fluorescence, indicating the occurrence of binding activity, with the mCherry fluorescence representing lentiviral infection in the control mouse incisor. However, the eGFP fluorescence was rarely detected in the *Arid1b* mutant mouse incisor (Response Figure 5C-5F). This *in vivo* evidence validates that ARID1B binds to the identified intron region of *Bcl11b*.

Response Figure 5. ARID1B binding activity validation *in vivo* using lentivirus injection. (A) Schematic diagram of the structure of Intronic Reporter Lentivirus. (B) Schematic diagram of the tamoxifen induction and lentivirus injection protocols. 5dpt, 5 days post-tamoxifen induction; 5dpi, 5 days post lentivirus injection. (C-F) Visualization of eGFP and mCherry fluorescence in control (C, E) and *Gli1-CreER;Arid1b^{fl/fl}* (D, F) mouse incisors. E and F represent high-magnification images of the boxes in C and D, respectively. Yellow arrow points to the co-localization of mCherry and eGFP signals. Scale bars, 100 μ m.

Minor comments:

1. The introduction could be shortened. There is too much review of every aspect covered in this manuscript. The authors need to decide what they want to cover and hit the high points.

We thank the reviewer for the suggestion. We revised and shortened the introduction.

2. In Figure 2F to J, the authors administer tamoxifen over a one month period. However, they do not state in the Materials and Methods or the Results section how this was accomplished. Was it once a day? Every other day? This needs to be explicitly stated at least in the Methods.

We thank the reviewer for the comment. To clarify, we administered tamoxifen by giving one injection per day for three consecutive days. We added this information to the Materials and Methods under the tamoxifen administration section.

3. The others need to state if the single cell experiments were performed one or two times. Current accepted procedures suggest that one time is sufficient when results are used as a tool for a study like this. However, this does need to be stated.

We thank the reviewer for the comment. We conducted a single scRNA-seq analysis for both the control and *Gli1-CreER;Ard1b^{fl/fl}* mouse incisors, employing a pooled sample of eight mouse incisors for each group. We added this information in the Materials and Methods under single-cell RNA sequencing and single-cell ATAC sequencing.

4. In Figure 4B, there needs to be some sort of heat map legend in the figure.

We thank the reviewer for the comment. We updated Figure 4B with the appropriate legend.

Once again, we want to thank the reviewers for all the constructive suggestions and feel that our revised manuscript has greatly improved through this review process. Thank you.

With best wishes,

Yang Chai, DDS, PhD
University Professor
George & MaryLou Boone Chair
in Craniofacial Molecular Biology
Center for Craniofacial Molecular Biology
University of Southern California
2250 Alcazar Street, CSA 103
Los Angeles, CA 90033
e-mail: ychai@usc.edu
phone: (323) 442-3480

Reviewer #1 (Remarks to the Author):

The authors have satisfactorily and reasonably addressed most of my concerns and comments

Reviewer #3 (Remarks to the Author):

In this revision by Zhang et al., the authors have done a very good job responding to all comments. In some cases, they have provided response figures, while at other points, they have modified existing figures to include new data. My comments are rather short and primarily address aspects that would help clarify approaches or interpretations. However, I do want to point out that while leaky Cre expression can occur, I do not believe it is as common as stated by Reviewer 2. VERY FEW papers utilizing Cres or Cre-ER approaches are required to prove the absence of leaky expression or compromised cell viability, due primarily to absence of strong data that this is a substantial problem.

1. The authors have responded with a "response figure" to reviewer 1's comment asking about the affect of ARID1B absence on the expression of ARID1A. The authors' show in the response Figure 1 that ARID1A expression goes up following the loss of ARID1B, suggesting that ARID1A potentially partially compensating for the loss of ARID1B. They further show an interesting phenotype when one copy of ARID1A is removed in an ARID1B mutant background. The implication is that if both A and B were removed completely, there could be massive cell proliferation, potentially affecting the ability to sustain incisor growth after birth. I am still not clear why the authors do not include this in the manuscript. Even if it is a supplemental figure referenced as part of the Discussion, it should be included (and discussed!).

2. The authors have now added details to their tamoxifen injections. The need to do the same for the EdU injections. They state on Page 7, "we injected control and Gli1-CreER;Arid1bfl/fl mice with EdU for a 1-month period beginning from postnatal day 5 and analyzed the cells after another month." Was that a single injection or multiple injections? The need to add the specific details here as they did for tamoxifen.

3. This is a small point, but the description of the authors' analysis of TGFb superfamily expression in their scRNA-seq/bulk RNA seq is a bit confusing because they do not provide "Supplementary Figure 7" soon enough. It seems they are simply listing what they are doing without providing data. The correct wording would be. "Our analysis did not reveal significant changes in BMP signaling "Supplementary Figure 7".

4. The authors should quantify their western analysis shown in Figures 6A and 7Q. While I agree there does appear to upregulation, beta-actin is also higher in the control sample (at least for Figure 6A). This can be put in as Supplemental data.

March 19, 2024

NCOMMS-23-37342A

Point-by-point response to the reviewers' comments:

Reviewer #1 (Remarks to the Author):

The authors have satisfactorily and reasonably addressed most of my concerns and comments

We appreciate the reviewer's valuable input for improving our manuscript and the positive feedback on our revision.

Reviewer #3 (Remarks to the Author):

In this revision by Zhang et al., the authors have done a very good job responding to all comments. In some cases, they have provided response figures, while at other points, they have modified existing figures to include new data. My comments are rather short and primarily address aspects that would help clarify approaches or interpretations. However, I do want to point out that while leaky Cre expression can occur, I do not believe it is as common as stated by Reviewer 2. VERY FEW papers utilizing Cres or Cre-ER approaches are required to prove the absence of leaky expression or compromised cell viability, due primarily to absence of strong data that this is a substantial problem.

We appreciate the reviewer's valuable input for improving our manuscript and the positive feedback on our revision.

1. The authors have responded with a "response figure" to reviewer 1's comment asking about the affect of ARID1B absence on the expression of ARID1A. The authors' show in the response Figure 1 that ARID1A expression goes up following the loss of ARID1B, suggesting that ARID1A potentially partially compensating for the loss of ARID1B. They further show an interesting phenotype when one copy of ARID1A is removed in an ARID1B mutant background. The implication is that if both A and B were removed completely, there could be massive cell proliferation, potentially affecting the ability to sustain incisor growth after birth. I am still not clear why the authors do not include this in the manuscript. Even if it is a supplemental figure referenced as part of the Discussion, it should be included (and discussed!).

We thank the reviewer for the comments. We included this data in the revised Supplementary Figure 11 and added further discussion of this data as highlighted in the text.

2. The authors have now added details to their tamoxifen injections. The need to do the same for the EdU injections. They state on Page 7, "we injected control and Gli1-CreER;Arid1bfl/fl mice with EdU for a 1-month period beginning from postnatal day 5 and analyzed the cells after

another month.” Was that a single injection or multiple injections? The need to add the specific details here as they did for tamoxifen.

We thank the reviewer for the comments. We added the corresponding EdU injection approach to the “EdU incorporation, staining, and TUNEL assays” section of the Methods.

3. This is a small point, but the description of the authors’ analysis of TGFb superfamily expression in their scRNA-seq/bulk RNA seq is a bit confusing because they do not provide “Supplementary Figure 7” soon enough. It seems they are simply listing what they are doing without providing data. The correct wording would be. “Our analysis did not reveal significant changes in BMP signaling “Supplementary Figure 7”.

We thank the reviewer for the comments. We included the bulk RNA-seq and scRNA-seq data to show the expression levels of the ligands and receptors belonging to the BMP signaling pathway within mesenchymal cells in the revised Supplementary Figure 7.

4. The authors should quantify their western analysis shown in Figures 6A and 7Q. While I agree there does appear to upregulation, beta-actin is also higher in the control sample (at least for Figure 6A). This can be put in as Supplemental data.

We thank the reviewer for the comments. We quantified the western blots and included the data in revised Supplementary Figure 9.